# Oral Combination Treatment of Gefitinib (Iressa™) and Sasam-Kyeongokgo: Synergistic Effects on the NCI-H520 Tumor Cell Line

**Jeong-Hoon Oh** [1,†], **Joo Wan Kim** [2,†], **Chul-Jong Jung** [3], **Jae-Suk Choi** [4,*] and **Sae Kwang Ku** [1,*]

1 Department of Histology and Anatomy, College of Korean Medicine, Daegu Haany University, Gyeongsan-si 38610, Republic of Korea
2 Department of Companion Animal Health, Daegu Haany University, Gyeongsan-si 38610, Republic of Korea
3 Research Institute, Okchundang Inc., Daegu 41059, Republic of Korea
4 Department of Seafood Science and Technology, The Institute of Marine Industry, Gyeongsang National University, 38 Cheondaegukchi-gil, Tongyeong-si 53064, Republic of Korea
* Correspondence: jsc1008@gnu.ac.kr (J.-S.C.); gucci200@hanmail.net (S.K.K.); Tel.: +82-55-772-9142 (J.-S.C.); +82-53-819-1549 (S.K.K.)
† These authors contributed equally to this work.

**Abstract:** The aim of this research was to confirm the possible synergic effects of sasam-Kyeongokgo (SKOG) on the anti-tumor activity of gefitinib using athymic nude mice bearing the human non-small-cell lung squamous cell carcinoma (NSCLC) NCI-H520 cell line after continuous oral combination treatment provided daily for 35 days within a timeframe of 5 min, as a developing process of novel preventive and therapeutic regimes for various types of lung cancer. After 35 days, BW, tumor volume and weight, and lymphatic and periovarian fat pad weight measurements, as well as serum IFN-γ and IL-6 level, NK cell activity, and splenic cytokine content measurements, together with histopathological observations, and immunohistochemistry measurements of the treated and control mice, were performed. The results suggest that the co-administration of SKOG 400, 200, or 100 mg/kg with gefitinib markedly increased the anti-tumor activity of gefitinib through immunostimulatory effects and also dramatically inhibited cancer cachexia through the immunostimulatory effect, potentiating the anti-tumor activity of gefitinib, with favorable anti-cachexia effects. Therefore, the co-administration of over 100 mg/kg of SKOG and gefitinib can provide an effective novel treatment regimen for various lung cancer patients.

**Keywords:** sasam-Kyeongokgo; gefitinib; synergic effects; lung cancer; anti-tumor activity; carcinoma; nude mice



## 1. Introduction

Gefitinib, a representative epidermal growth factor receptor (EGFR) inhibitor, is an oral anticancer drug that is frequently used to treat various malignancies, such as lung and breast cancer, but its use is limited by concerns about its hepatotoxicity [1,2]. The drug sasam, which consists of the dried roots of *Adenophora triphylla* var *japonica* (Adenophora, Campanulaceae), has traditionally been utilized as a herbal medicine for its anti-inflammatory, hepatoprotective [3], and antitussive [4] effects in Korea, China, and Japan [5]. Various essential oils, triterpenoids, and alkaloids, such as piperidine, pyrrolidine, heptacosane, triphyllol, lupenone, nonacosane, and saponine, have been isolated and purified from sasam [3–6], and their anti-obesity [5,6], anticancer [7], hepatoprotective [4], mucus production (expectoration) [8], and antitussive properties [9] have been experimentally demonstrated. Kyeongokgo is a prescription of the *Donguibogam*, a Korean traditional medicine book, consisting of *Panax ginseng* (Araliaceae), *Rehmannia glutinosa* (Orobanchaceae), *Poria cocos* (Polyporaceae), and honey. Kyeongokgo is a traditional prescription that has been used

to achieve health and longevity in middle-aged and elderly people in Korea since ancient times [10,11].

Thus far, Kyeongokgo has been experimentally proven to show various pharmacological effects, including anti-hyperlipidemic [10], antioxidant [11], anti-inflammatory [12], immunomodulatory [11], osteoporosis-treating [13], hair-growth-promoting [14], anti-fatigue and athletic-performance-improving [15], growth-promoting [16], anti-aging [17], and antibacterial effects [18]. In addition, we have already evaluated the respiratory function improvement effects of Adenophorae Radix [9], Kyeongokgo [19], and Sasam-Kyeongokgo [20] through in vivo animal experiments. Therefore, an appropriate combination of sasam-Kyeongokgo and gefitinib is expected to synergistically increase the anticancer effect of gefitinib on lung cancer.

In this study, as part of the effort to develop novel anticancer therapies for lung cancer patients, the effects of sasam-Kyeongokgo on the anticancer activity of gefitinib were evaluated in vitro using human NSCLC, NCI-H520 cells. Subsequently, after the oral administration of sasam-Kyeongokgo to athymic nude mice transplanted with the NCI-H520 lung cancer cell line, indices related to anticancer and immune activation effects, such as the tumor and lymph organ weight and histopathological changes, as well as the levels of cytokines in the blood and immune organs, were monitored.

## 2. Materials and Methods

### 2.1. Test Substance and Oral Administration

Black-brown viscous sasam-Kyeongokgo was provided by Okchundang (Gyeongsan-si, Republic of Korea). Part of the test material was stored at the Medical Research Center (Daegu Haany University, Gyeongsan-si, Republic of Korea, code No. SKOG2017Ku01).

For the production of sasam-Kyeongokgo, 4500 g of dried Panax ginseng powder, 9000 g of Poria cocos powder, 4500 g of dried powder of A. triphylla var japonica roots, 39,000 g of honey, and 47,000 g of fresh juice of Rehmannia glutinosa were mixed with a stirrer at room temperature and transferred to a hot water container, and the outer perimeter of the container was filled to the level of approximately 70% with water. The water in the water bath was heated, and the temperature was kept constant at 80 °C for 72 h, after which the water bath was naturally cooled until it reached room temperature, ensuring that the sasam-Kyeongokgo had undergone secondary heating (80 °C) and the natural cooling process. On the basis of a previously published UPLC analysis [20], the sasam-Kyeongokgo used in this experiment was considered to contain lupeol (224.52 $\pm$ 12.5 mg/kg), syringaldehyde (0.142 $\pm$ 0.014 mg/kg), 5-hydroxymethyl-2-furaldehyde (5H2F; 559.50 $\pm$ 1.70 mg/kg), acteoside (0.309 $\pm$ 0.008 mg/kg), catalposide (0.328 $\pm$ 0.01 mg/kg), and Rg3 (4.418 $\pm$ 0.02 mg/kg). Pale yellow gefitinib powder was obtained from Suzhou Huihe Pharm Co., Ltd. (Suzhou, China) and used after being dissolved in sterile distilled water.

### 2.2. In Vitro Assessment of Anticancer Effects

After the treatment of the NCI-H520 cells ($1 \times 10^4$ cells) with sasam-Kyeongokgo (0, 0.01, 0.1, 0.5, 1, 5, 10, and 40 mg/kg) and gefitinib (0, 0.001, 0.01, 0.1, 1, 5, 10, and 50 μM), the concentration that inhibited the survival rate by 50%, $IC_{50}$, was evaluated using the general MTT method.

### 2.3. In Vivo Evaluation of Anticancer Effects

#### 2.3.1. Animals

A total of 56 SPF/VAF Hsd:Athymic Nude-Foxn1nu mice (6-week-old female mice, Harlan Lab., Udine, Italy) were obtained and designated based on the BW range after 7 days of acclimatization. The NCI-H520 cells were transplanted into the subcutaneous area of the right hip of the rodents. At 14 days after tumor cell transplantation, the transplanted mice with a tumor volume of 500.14 $\pm$ 125.31 mm$^3$ (314.87~771.33 mm$^3$) or more were selected, and eight mice per group were used in this experiment. An additional eight intact controls were also prepared based on BW (intact group: mean weight, 21.34 $\pm$ 1.20 g;

weight range, 19.80~23.20 g; tumor transplant group: mean weight, 21.35 ± 1.15 g; weight range, 19.40–23.60 g). These animal studies were performed with the approval of the Animal Experimentation Ethics Committee of Daegu Haany University (Gyeongsan-si, Republic of Korea) (approval no. DHU2017-098).

2.3.2. Dosing and Definitions of the Study Groups

In this experiment, the gefitinib dose was set as 120 mg/kg, which has previously shown anticancer effects on mice with xenografts [21], while sasam-Kyeongokgo doses of 400, 200, and 100 mg/kg and an administration interval of 5 min between gefitinib and sasam-Kyeongokgo were set on the basis of the results of previous animal experiments [20].

The animals were placed in the following seven groups (eight animals per group):

(1)  Intact control: animals that received the normal vehicle (sterile distilled water: 10 mL/kg) alone.
(2)  Tumor-bearing control: animals that received sterile distilled water after NCI-H520 tumor cell transplantation.
(3)  G120 group: animals that received gefitinib (120 mg/kg) alone after tumor cell transplantation.
(4)  SKOG400: animals that received sasam-Kyeongokgo (400 mg/kg) alone after tumor cell transplantation.
(5)  GSK400: animals that received gefitinib (120 mg/kg) and sasam-Kyeongokgo (400 mg/kg) after tumor cell transplantation.
(6)  GSK200: animals that received gefitinib (120 mg/kg) and sasam-Kyeongokgo (200 mg/kg) after tumor cell transplantation.
(7)  GSK100: animals that received gefitinib (120 mg/kg) and sasam-Kyeongokgo (100 mg/kg) after tumor cell transplantation.

For the tumor cell transplantation, NCI-H520 (ATCCC, American Type Culture Collection Center, Manassas, VA, USA) cells were cultured in RPMI 1640 (Gibco, Grand Island, NY, USA) medium supplemented with 10% fetal bovine serum (FBS) and maintained by subculture in an incubator (Model 311; Thermo Forma, Marietta, OH, USA) at 37 °C in 5% $CO_2$. A tumor cell suspension was prepared at a concentration of $1.0 \times 10^8$ cells/mL, and 0.2 mL ($2 \times 10^7$ cells/mouse) of the suspension was transplanted subcutaneously into the right dorsal buttock of each mouse to induce the formation of a solid tumor mass.

Fifteen days after the transplantation of the NCI-H520 lung cancer cells into the nude mice, sasam-Kyeongokgo was suspended in sterile distilled water, and concomitant oral administration of the suspension was performed for 35 days at doses of 400, 200, and 100 mg/kg of BW once daily within 5 min after the administration of gefitinib, which was orally administered at a dose of 120 mg/kg.

In addition, the same dose of sterile DW was provided, instead of the second agent, to groups receiving the sasam-Kyeongokgo and gefitinib single administration. For the normal and tumor transplantation medium control groups, sterile distilled water alone was administered twice at intervals of 5 min.

2.3.3. Observation Items

The anticancer and immune activation effects, as well as the tumor-related cachexia, in mice transplanted with the NCI-H520 lung cancer cell line were evaluated by measuring the parameters described below:

(1)  Anticancer effect: the tumor volume, tumor weight, changes in the tumor cell volume, apoptotic cell percentage in the formed mass, and changes in the cyclo-oxygenase-2 (COX-2), caspase-3, poly (ADP-ribose) polymerase (PARP), tumor necrosis factor (TNF)-$\alpha$, and inducible nitric oxide synthase (iNOS) immunoreactivity in the tumor mass.
(2)  Immune activation effect: the immune organ (thymus and submandibular lymph node) weight, blood interferon (IFN)-$\gamma$ content, natural killer (NK) cell activity, and spleen TNF-$\alpha$, interleukin (IL)-10, and IL-1$\beta$ content changes, as well as histological

changes in the immune organs and changes in the TNF-$\alpha$ immunoreactivity of the tumor mass and submandibular lymph nodes.

(3) Tumor-related cachexia-inhibiting effect: changes in the BW, periovarian fat weight, and blood IL-6 content and histological changes in the periosteal fat.

The experimental methods and related observation items are presented in the Supplementary Materials.

The selection, analysis, and measurement of biomarkers related to the anticancer effect and the immune activation effect in this study were based on the methods suggested by previous researchers [22–26]. The selection, analysis, and measurement of biomarkers related to the tumor-related cachexia inhibitory effect in this study were based on the methods described by Park et al. (2014), Iizuka et al. (2000), and Kurebayashi et al. (1999) [22,27,28]. The primary antisera and detection kits utilized in this study are shown in Appendix A.

### 2.4. Statistical Analyses

Using the SPSS program (14.0 K, SPSS Inc., Chicago, IL, USA), statistical and Probit analyses were carried out [22,29] in accordance with the protocols described in our previous study [30].

## 3. Results

### 3.1. In Vitro Cytotoxicity

Compared with the vehicle group (0 mg/mL treatment group), a significant decrease ($p < 0.01$) in the viability of the NCI-H520 cells was confirmed, starting with the sasam-Kyeongokgo 1 mg/mL treatment group and the gefitinib 0.1 μM treatment group, and the IC$_{50}$ values were 17.21 $\pm$ 7.28 mg/mL (Figure 1a) and 4.56 $\pm$ 1.46 μM (2.00 $\pm$ 0.64 μg/mL) (Figure 1b), respectively.

### 3.2. Changes in BW and Weight Gain

The tumor transplantation control group showed no significant change in BW (actual BW–BW at initial administration) compared with the intact control group during the entire experimental period. However, in comparison with the intact control group, the tumor transplantation control group exhibited a significant decrease ($p < 0.01$) in the actual BW, excluding the tumor weight (BW at sacrifice–tumor weight at sacrifice), and a decrease in the level of gain during the administration period. In the gefitinib-alone group, significant ($p < 0.01$) weight loss was observed 28 days after the initiation of administration in comparison with the tumor transplantation control group. In comparison with the tumor transplantation control group, significant actual BW gain and a decrease in the level of gain during the administration period based on BW were also confirmed ($p < 0.01$ or $p < 0.05$) in the gefitinib-alone group. In contrast, a significant increase ($p < 0.01$ or $p < 0.05$) in the actual BW and gain was confirmed in the group administered sasam-Kyeongokgo alone and the groups administered combined gefitinib and sasam-Kyeongokgo (400, 200, and 100 mg/kg) in comparison with the tumor transplantation control group. In particular, a significant increase ($p < 0.01$) in BW was observed in the group administered gefitinib (120 mg/kg) and sasam-Kyeongokgo (400, 200, and 100 mg/kg) in comparison with the group administered gefitinib (120 mg/kg) alone 28 days after the initiation of administration, and a significant increase ($p < 0.01$) in the actual BW and gain was also observed in comparison with the group administered gefitinib alone (Table 1, Figure 2).

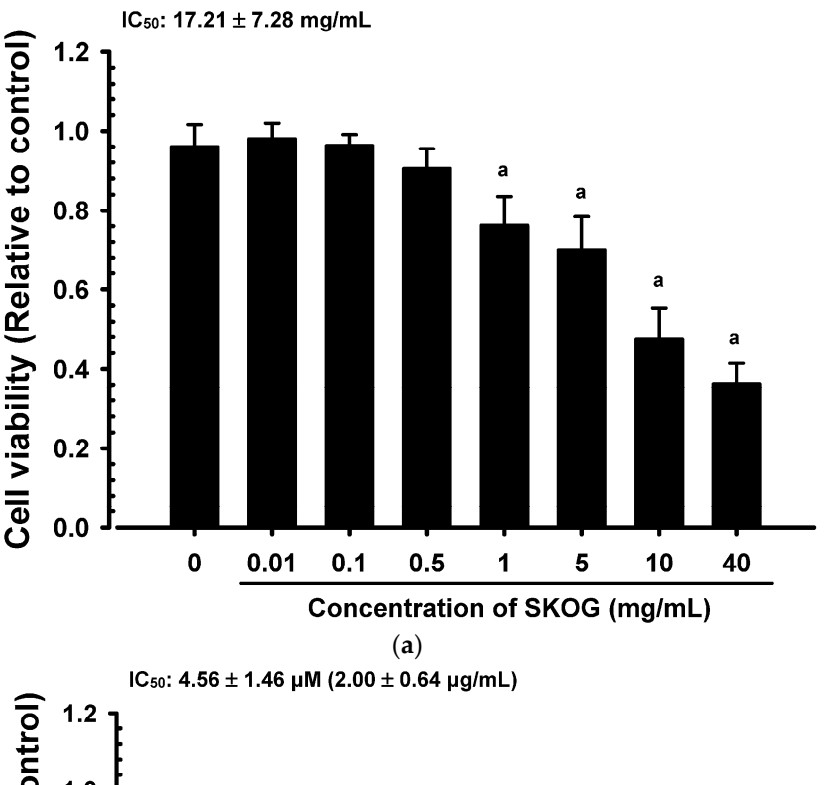

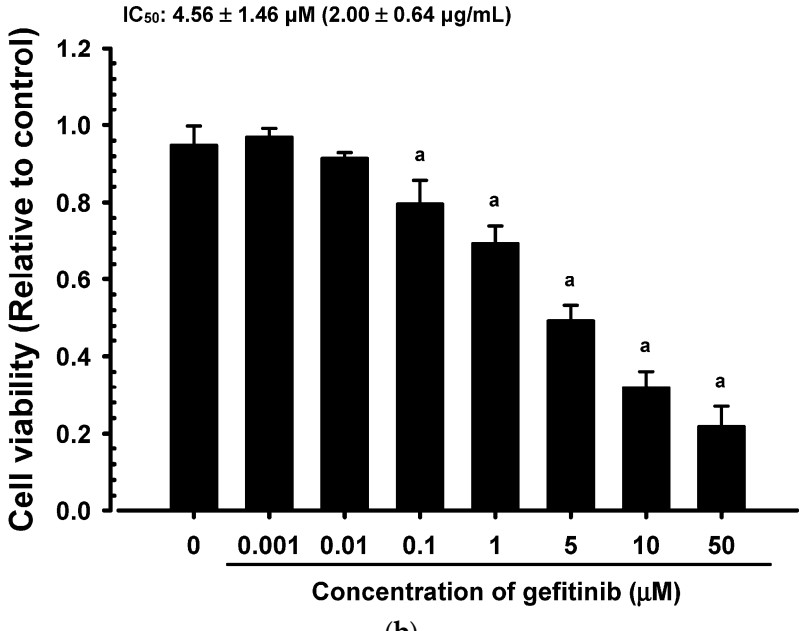

**Figure 1.** (**a**) Effects of SKOG on the NCI-H520 tumor cell viability. AR = Adenophorae Radix; KOG = *Kyeongokgo*; SKOG = *Sasam-Kyeongokgo*. IC$_{50}$ = 50% inhibitory concentration. [a] $p < 0.01$ versus control (0 mg/mL), assessed by the Mann–Whitney U test. (**b**) Effects of gefitinib on the NCI-H520 tumor cell viability. IC$_{50}$ = 50% inhibitory concentration. [a] $p < 0.01$ versus control (0 µM), assessed by the Mann–Whitney U test.

**Table 1.** Changes in BW gain in NCI-H520 tumor cell xenograft mice.

| Groups | BWs (g) | | | BW Gains (g) |
|---|---|---|---|---|
| | At First Administration | At Sacrifice | Actual BWs | |
| Controls | | | | |
| Intact | 19.45 ± 1.31 | 23.04 ± 0.88 | 23.04 ± 0.88 | 3.59 ± 0.56 |
| TB | 19.43 ± 1.24 | 22.58 ± 0.47 | 19.77 ± 0.63 [a] | 0.34 ± 0.90 [a] |

**Table 1.** *Cont.*

| Groups | BWs (g) | | | BW Gains (g) |
|---|---|---|---|---|
| | At First Administration | At Sacrifice | Actual BWs | |
| Single-formula-treated | | | | |
| Gefitinib | 19.29 ± 1.29 | 20.13 ± 0.76 [fg] | 18.46 ± 0.77 [ad] | −0.83 ± 0.98 [ac] |
| SKOG | 19.38 ± 1.39 | 23.03 ± 1.65 [gh] | 20.99 ± 1.59 [ade] | 1.61 ± 0.67 [ace] |
| Gefitinib and SKOG | | | | |
| 400 mg/kg | 19.33 ± 1.34 | 23.40 ± 1.37 [h] | 22.93 ± 1.44 [ce] | 3.60 ± 0.71 [ce] |
| 200 mg/kg | 19.23 ± 1.41 | 22.65 ± 1.42 [h] | 21.94 ± 1.42 [ce] | 2.72 ± 0.45 [bc] |
| 100 mg/kg | 19.20 ± 0.64 | 22.45 ± 0.85 [h] | 21.19 ± 0.90 [ade] | 1.99 ± 0.65 [ace] |

TB = tumor-bearig; AR = Adenophorae Radix; KOG = *Kyeongokgo*; SKOG = *sasam-Kyeongokgo*. [a] $p < 0.01$ and [b] $p < 0.05$ versus intact control, assessed by the least significant difference (LSD) multi-comparison test; [c] $p < 0.01$ and [d] $p < 0.05$ versus TB control, assessed by the least significant difference (LSD) multi-comparison test; [e] $p < 0.01$ versus gefitinib-single-formula-treated rodents, assessed by the least significant difference (LSD) multi-comparison test; [f] $p < 0.01$ versus intact control, asessed by the Mann–Whitney U test; [g] $p < 0.01$ versus TB control, assessed by the Mann–Whitney U test; [h] $p < 0.01$ versus gefitinib-single-formula-treated rodents, assessed by the Mann–Whitney U test.

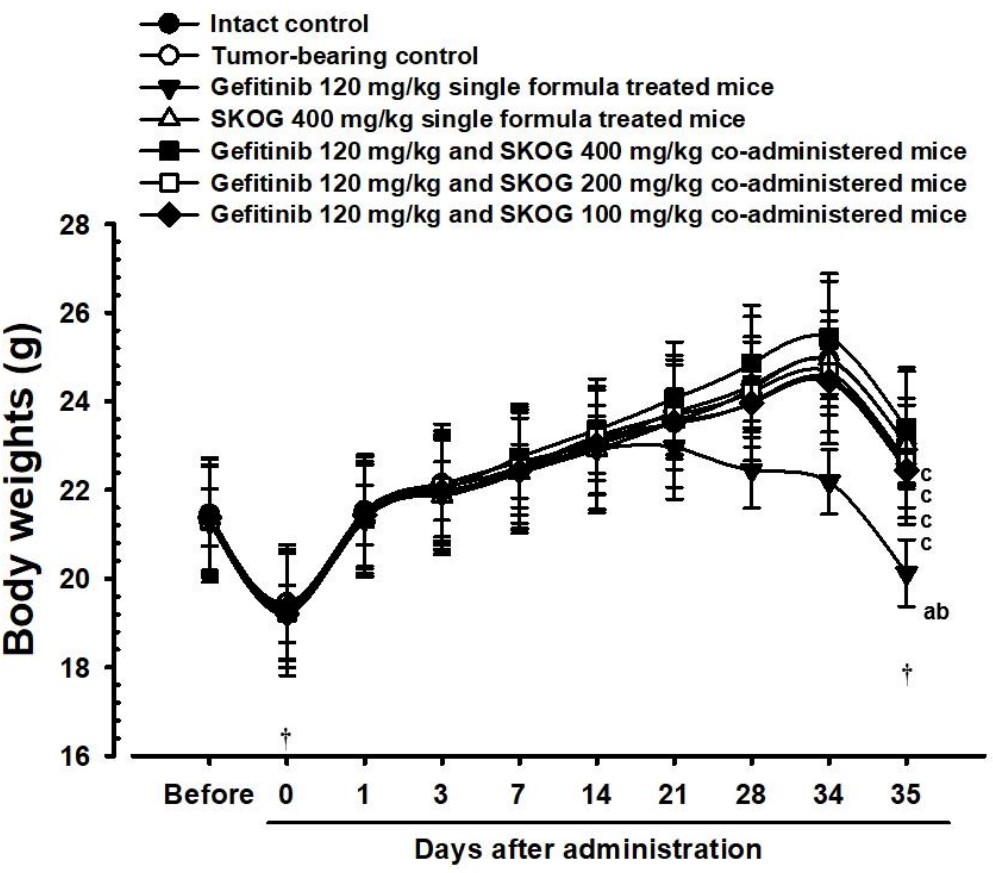

**Figure 2.** BW changes in NCI-H520 tumor cell xenograft mice. AR = Adenophorae Radix; KOG = *Kyeongokgo*; SKOG = *sasam-Kyeongokgo*. Before means 1 day before the initiation of administration at day 13 day after tumor implantation. Before sacrifice and the day of initiation of the treatment, all the rodents were overnight fasted (†). [a] $p < 0.01$ versus intact control assessed by the Mann–Whitney U test; [b] $p < 0.01$ versus TB control, assessed by the Mann–Whitney U test; [c] $p < 0.01$ versus gefitinib-single-formula-treated rodents, assessed by the Mann–Whitney U test.

### 3.3. Changes in the Tumor Volume

The gefitinib-alone group showed significant reductions ($p < 0.01$) in the tumor volume 14 days after the initiation of administration, as well as in the level of change in the tumor

volume during the administration period, in comparison with the tumor transplantation control group. The group treated with 400 mg/kg sasam-Kyeongokgo alone also showed significant ($p < 0.01$) reductions in the tumor volume 21 days after the initiation of administration, as well as in the level of change in the tumor volume during the administration period, in comparison with the tumor transplantation control group. Similarly, the groups administered sasam-Kyeongokgo and gefitinib confirmed significant reductions ($p < 0.01$ or $p < 0.05$) in the tumor volume 14 days after the initiation of administration and in the level of change in the tumor volume during the administration period in comparison with the tumor transplantation control group ($p < 0.01$). In comparison with the group administered gefitinib alone, the group that received a combination of 400 mg/kg sasam-Kyeongokgo and gefitinib showed a decrease in the tumor volume 14 days after the initiation of administration, while the group administered 200 mg/kg sasam-Kyeongokgo and gefitinib showed a decrease in the tumor volume 21 days after the initiation of administration ($p < 0.01$ or $p < 0.05$). Additionally, the level of change in the tumor volume during the administration period also showed a significant decrease ($p < 0.01$) in the 400 mg/kg sasam-Kyeongokgo and gefitinib combination groups in comparison with the gefitinib-alone group (Table 2, Figures 3 and 4).

**Table 2.** Changes in the tumor volume in NCI-H520 tumor cell xenograft mice.

| Groups | Tumor Volume (mm$^3$) | | Changes (mm$^3$) (B–A) |
|---|---|---|---|
| | **First Administration (A)** | **Sacrifice (B)** | |
| Control | | | |
|    TB | 469.02 ± 102.47 | 6487.36 ± 862.66 | 6018.34 ± 774.83 |
| Single formula treated | | | |
|    Gefitinib | 462.10 ± 105.87 | 2858.51 ± 377.01 [a] | 2396.41 ± 329.84 [a] |
|    SKOG | 472.89 ± 141.15 | 3972.27 ± 655.53 [ab] | 3499.37 ± 557.32 [ab] |
| Gefitinib and SKOG | | | |
|    400 mg/kg | 473.39 ± 137.02 | 1209.50 ± 335.04 [ab] | 736.11 ± 235.75 [ab] |
|    200 mg/kg | 477.76 ± 161.99 | 1670.83 ± 471.94 [ab] | 1193.07 ± 375.71 [ab] |
|    100 mg/kg | 475.73 ± 122.90 | 2125.75 ± 442.71 [ab] | 1650.02 ± 431.10 [ab] |

TB = tumor-bearing; AR = Adenophorae Radix; KOG = *Kyeongokgo*; SKOG = *sasam-Kyeongokgo*. Before the first administration and sacrifice, all the rodents were fasted overnight. [a] $p < 0.01$ versus TB control, assessed by the Mann–Whitney U test; [b] $p < 0.01$ versus gefitinib-single-formula-treated rodents, assessed by the Mann–Whitney U test.

### 3.4. Changes in the Tumor Weight

Significant reductions ($p < 0.01$) in the tumor absolute and relative weights were confirmed in all the drug administration groups in comparison with the tumor transplantation control group. In particular, significant reductions in the tumor relative and absolute weights were confirmed in all the sasam-Kyeongokgo and gefitinib combination groups ($p < 0.01$ or $p < 0.05$), even compared with the gefitinib-alone group (Tables 3 and 4).

**Table 3.** Changes in the absolute tumor mass and organ weights in the NCI-H520 tumor cell xenograft mice.

| Groups | Tumor Mass | Spleen | Submandibular Lymph Node | Periovarian Fat Pad |
|---|---|---|---|---|
| Controls | | | | |
|    Intact | | 0.110 ± 0.011 | 0.013 ± 0.004 | 0.050 ± 0.013 |
|    TB | 2.808 ± 0.282 | 0.054 ± 0.005 [a] | 0.004 ± 0.001 [d] | 0.008 ± 0.002 [d] |
| Single-formula-treated | | | | |
|    Gefitinib | 1.664 ± 0.292 [f] | 0.051 ± 0.007 [a] | 0.004 ± 0.001 [d] | 0.007 ± 0.002 [d] |
|    SKOG | 2.036 ± 0.249 [f] | 0.080 ± 0.008 [abc] | 0.008 ± 0.002 [dfg] | 0.024 ± 0.004 [dfg] |

**Table 3.** *Cont.*

| Groups | Tumor Mass | Spleen | Submandibular Lymph Node | Periovarian Fat Pad |
|---|---|---|---|---|
| Gefitinib and SKOG | | | | |
| 400 mg/kg | 0.472 ± 0.130 [fg] | 0.103 ± 0.011 [bc] | 0.011 ± 0.002 [fg] | 0.033 ± 0.007 [dfg] |
| 200 mg/kg | 0.706 ± 0.278 [fg] | 0.097 ± 0.008 [abc] | 0.010 ± 0.002 [fg] | 0.028 ± 0.006 [dfg] |
| 100 mg/kg | 1.260 ± 0.198 [fh] | 0.082 ± 0.012 [abc] | 0.009 ± 0.001 [efg] | 0.024 ± 0.005 [dfg] |

TB = tumor-bearing; AR = Adenophorae Radix; KOG = *Kyeongokgo*; SKOG = *sasam-Kyeongokgo*; [a] $p < 0.01$ versus intact control, assessed by the least significant difference (LSD) multi-comparison test; [b] $p < 0.01$ versus TB control, assessed by the by the least significant difference (LSD) multi-comparison test; [c] $p < 0.01$ versus gefitinib-single-formula-treated rodents, assessed by the least significant difference (LSD) multi-comparison test; [d] $p < 0.01$ and [e] $p < 0.05$ versus intact control, assessed by the Mann–Whitney U test; [f] $p < 0.01$ versus TB control, assessed by the Mann–Whitney U test; [g] $p < 0.01$ and [h] $p < 0.05$ versus gefitinib-single-formula-treated rodents, assessed by the Mann–Whitney U test.

**Table 4.** Changes in the relative tumor mass and organ weights in the NCI-H520 tumor cell xenograft mice.

| Groups | Tumor Mass | Spleen | Submandibular Lymph Node | Periovarian Fat Pad |
|---|---|---|---|---|
| Controls | | | | |
| Intact | | 0.479 ± 0.060 | 0.057 ± 0.017 | 0.216 ± 0.051 |
| TB | 12.453 ± 1.378 | 0.241 ± 0.028 [a] | 0.019 ± 0.007 [e] | 0.037 ± 0.008 [e] |
| Single-formula-treated | | | | |
| Gefitinib | 8.274 ± 1.443 [g] | 0.251 ± 0.036 [a] | 0.019 ± 0.006 [e] | 0.036 ± 0.011 [e] |
| SKOG | 8.857 ± 1.042 [g] | 0.347 ± 0.043 [acd] | 0.034 ± 0.009 [egh] | 0.104 ± 0.020 [egh] |
| Gefitinib and SKOG | | | | |
| 400 mg/kg | 2.037 ± 0.632 [gh] | 0.440 ± 0.050 [cd] | 0.048 ± 0.010 [gh] | 0.140 ± 0.035 [egh] |
| 200 mg/kg | 3.123 ± 1.208 [gh] | 0.430 ± 0.029 [bcd] | 0.044 ± 0.005 [gh] | 0.125 ± 0.031 [egh] |
| 100 mg/kg | 5.623 ± 0.961 [gh] | 0.365 ± 0.042 [acd] | 0.039 ± 0.005 [fgh] | 0.105 ± 0.020 [egh] |

TB = tumor-bearing; AR = Adenophorae Radix; KOG = *Kyeongokgo*; SKOG = *sasam-Kyeongokgo*. Gefitinib (120 mg/kg) or SKOG (400 mg/kg) was administered to the single-formula-treated groups. [a] $p < 0.01$ and [b] $p < 0.05$ versus intact control, assessed by the least significant differences (LSD) multi-comparison test; [c] $p < 0.01$ versus TB control, assessed by the least significant differences (LSD) multi-comparison test; [d] $p < 0.01$ versus gefitinib-single-formula-treated rodents, assessed by the least significant difference (LSD) multi-comparison test; [e] $p < 0.01$ and [f] $p < 0.05$ versus intact control, assessed by the Mann–Whitney U test; [g] $p < 0.01$ versus TB control, assessed by the Mann–Whitney U test; [h] $p < 0.01$ versus gefitinib-single-formula-treated rodents by Mann–Whitney U test.

### 3.5. Changes in the Spleen Weight

The tumor transplantation control group showed a significant decrease ($p < 0.01$) in the absolute and relative weights of the spleen in comparison with the intact control group. A significant increase ($p < 0.01$) in the spleen weight was confirmed in the sasam-Kyeongokgo-alone group and sasam-Kyeongokgo and gefitinib combination groups, respectively, compared with the tumor transplantation control group. In particular, significant increases in the relative and absolute weights of the spleen were confirmed in all the sasam-Kyeongokgo and gefitinib combination groups compared with the gefitinib-alone group ($p < 0.01$). In contrast, no significant changes in the spleen absolute and relative weights were confirmed in the gefitinib-alone group compared with the tumor transplantation control group (Tables 3 and 4).

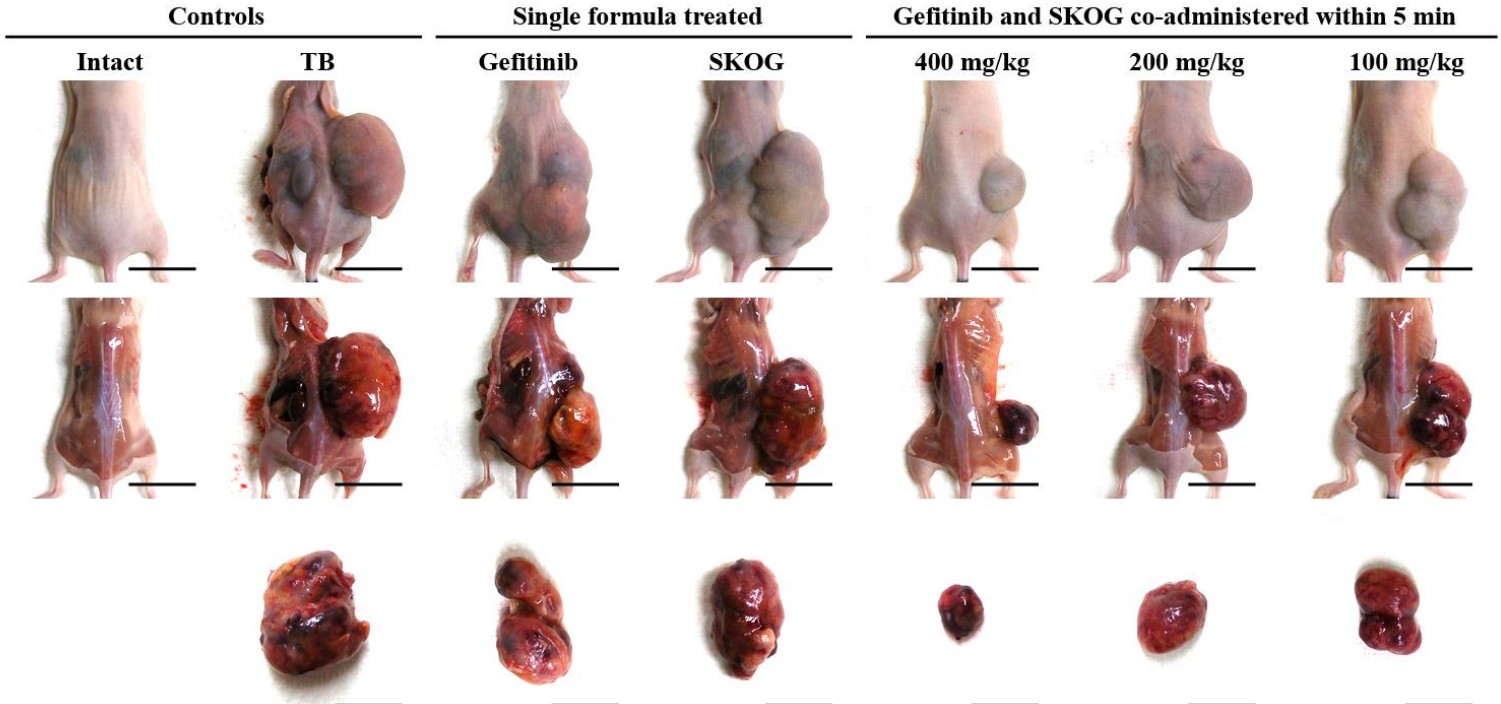

**Figure 3.** Representative gross tumor mass images of intact or NCI-H520 tumor cell xenograft mice taken at sacrifice. TB = tumor-bearing; AR = Adenophorae Radix; KOG = *Kyeongokgo*; SKOG = *sasam-Kyeongokgo*.

### 3.6. Changes in the Submandibular Lymph Node Weight

A significant decrease ($p < 0.01$) in the relative and absolute weights of the submandibular lymph nodes was observed in the tumor transplantation control group in comparison with the intact control group. However, a significant increase ($p < 0.01$) in the submandibular lymph node weight was confirmed in the group administered sasam-Kyeongokgo alone and the groups administered sasam-Kyeongokgo and gefitinib in comparison with the tumor transplantation control group. In particular, a significant increase ($p < 0.01$) in the relative and absolute weights of the submandibular lymph nodes was confirmed in the groups administered sasam-Kyeongokgo and gefitinib in comparison with the group administered gefitinib alone. However, no significant change was observed in the weight of the maxillary lymph nodes in the gefitinib-alone group in comparison with the tumor transplantation control group (Tables 3 and 4).

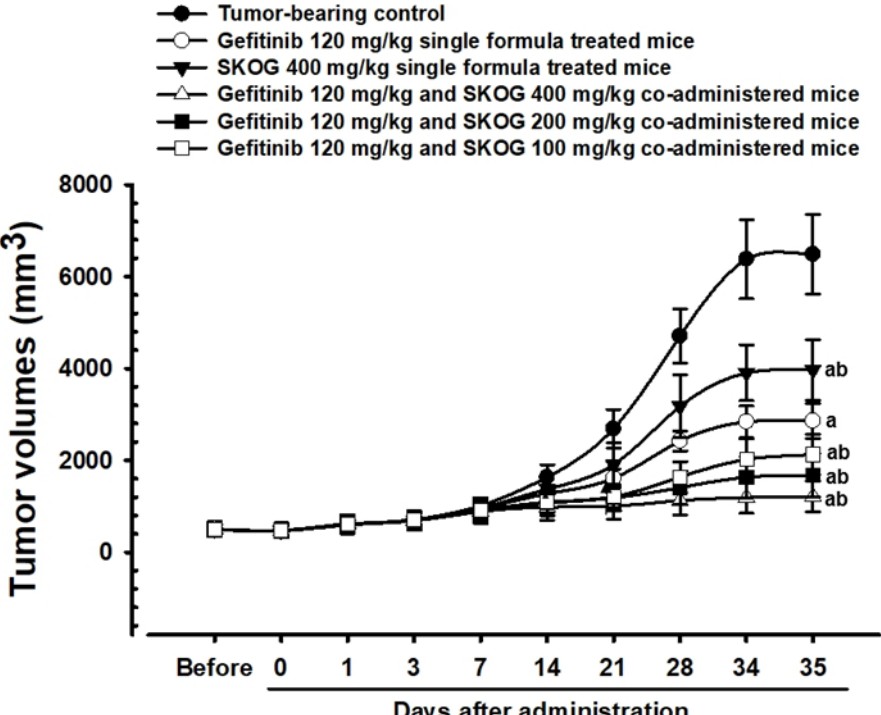

**Figure 4.** Tumor volume changes in NCI-H520 tumor cell xenograft mice. TB = tumor-bearing; AR = Adenophorae Radix; KOG = *Kyeongokgo*; SKOG = *sasam-Kyeongokgo*. Before means 1 day before the initiation of administration at day 13 day after tumor implantation. [a] $p < 0.01$ and [b] $p < 0.05$ versus TB control, assessed by the Mann–Whitney U test.

### 3.7. Changes in the Periovarian Fat Weight

Significant decreased ($p < 0.01$) in the relative and absolute periovarian fat weights were confirmed in the tumor transplantation control group in comparison with the intact control group. However, a significant increase ($p < 0.01$) in the periovarian fat weight was confirmed in the group administered sasam-Kyeongokgo alone and in the groups receiving sasam-Kyeongokgo with gefitinib at all three doses in comparison with the tumor transplantation control group. In particular, a significant increase ($p < 0.01$) in the periovarian fat weight was confirmed in the groups administered sasam-Kyeongokgo and gefitinib in comparison with the group administered gefitinib alone. On the other hand, no significant changes in the relative and absolute weights of periovarian fat were observed in the gefitinib-alone group in comparison with the tumor transplantation control group (Tables 3 and 4).

### 3.8. Changes in the IL-6 and IFN-γ Contents in the Blood

In the tumor transplantation control group, a significant increase ($p < 0.01$) in the serum IL-6 content and a decrease in the IFN-γ content were confirmed in comparison with the intact control group. A significant decrease ($p < 0.01$) in the serum IL-6 content and increase in the IFN-γ content were observed in the group administered sasam-Kyeongokgo alone or in combination with gefitinib in comparison with the tumor transplantation control group. In particular, a significant decrease ($p < 0.01$) in the serum IL-6 content and an increase in the IFN-γ content were confirmed in the groups administered sasam-Kyeongokgo and gefitinib at all three doses in comparison with the group administered gefitinib alone. In contrast, no significant changes in the serum IFN-γ and IL-6 levels were observed in the gefitinib-alone group in comparison with the tumor transplantation control group (Figure 5).

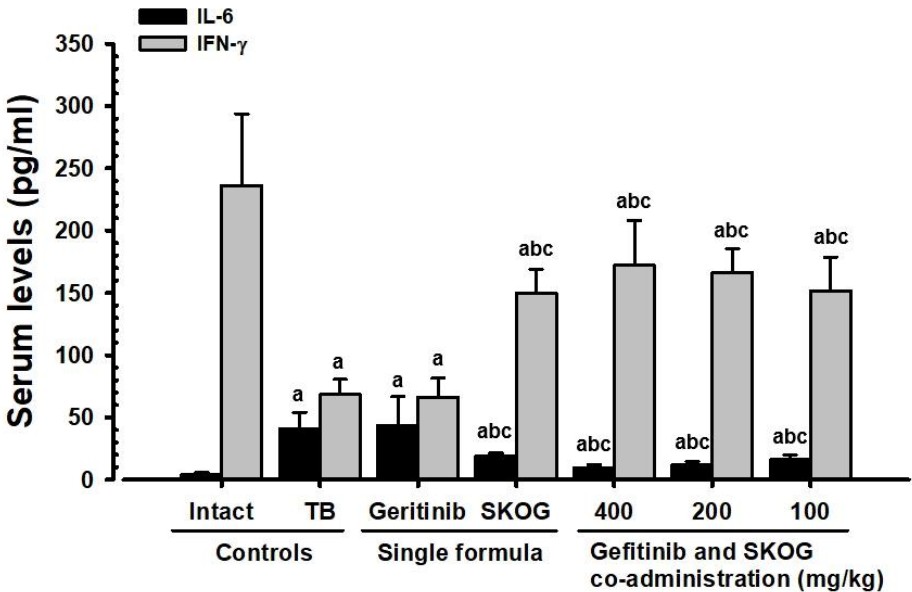

**Figure 5.** Serum IL-6 and IFN-γ level changes in NCI-H520 tumor cell xenograft mice. AR = Adenophorae Radix; KOG = *Kyeongokgo*; SKOG = *sasam-Kyeongokgo*; TB = tumor-bearing; IL = interleukin; IFN = interferon. [a] $p < 0.01$ versus intact control, assessed by the Mann–Whitney U test; [b] $p < 0.01$ versus TB control, assessed by the Mann–Whitney U test; [c] $p < 0.01$ versus gefitinib-single-formula-treated rodents, assessed by the Mann–Whitney U test.

### 3.9. Changes in NK Cell Activity

In the tumor transplantation control group, a significant decrease ($p < 0.01$) in the splenic and abdominal NK cell activity was observed in comparison with the intact control group. However, a significant increase ($p < 0.01$) in the splenic and abdominal NK cell activity was confirmed in the sasam-Kyeongokgo-alone group and in all the combined administration groups in comparison with the tumor transplantation control group. In particular, a significant increase ($p < 0.01$) in the splenic and abdominal NK cell activity was confirmed in the group administered sasam-Kyeongokgo and gefitinib in comparison with the group administered gefitinib alone. In contrast, no significant changes in the splenic and abdominal NK cell activity were observed in the gefitinib-alone group in comparison with the tumor transplantation control group (Figure 6).

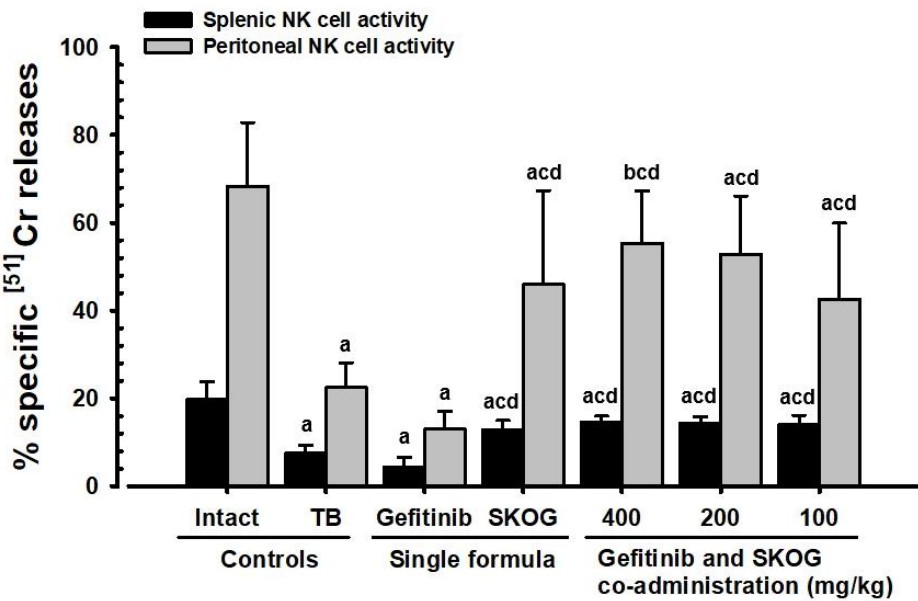

**Figure 6.** NK cell activity changes in NCI-H520 tumor cell xenograft mice. AR = Adenophorae Radix; KOG = *Kyeongokgo*; SKOG = *sasam-Kyeongokgo*; TB = tumor-bearing; NK = natural killer (macrophage). [a] $p < 0.01$ and [b] $p < 0.05$ versus intact control, assessed by the Mann–Whitney U test; [c] $p < 0.01$ versus TB control, assessed by the Mann–Whitney U test; [d] $p < 0.01$ versus gefitinib-single-formula-treated rodents, assessed by the Mann–Whitney U test.

*3.10. Changes in Spleen Cytokine Content*

In the tumor transplantation control group, a significant decrease ($p < 0.01$) in the splenic IL-1β, TNF-α, and IL-10 levels was confirmed in comparison with the intact control group. However, a significant increase ($p < 0.01$) in the spleen cytokine content was confirmed in all the sasam-Kyeongokgo-alone and combined administration groups compared with the tumor transplantation control group. In particular, a significant increase ($p < 0.01$) in the spleen TNF-α, IL-1β, and IL-10 levels was confirmed in the group administered sasam-Kyeongokgo and gefitinib in comparison with the group administered gefitinib alone. In contrast, no significant change in the splenic cytokine content was confirmed in the gefitinib-alone group in comparison with the tumor transplantation control group (Table 5).

**Table 5.** Changes in the splenic cytokine contents in NCI-H520 tumor cell xenograft mice.

| Groups | Tumor Necrosis Factor-α | Interleukin-1β | Interleukin-10 |
|---|---|---|---|
| Controls | | | |
| Intact | 142.83 ± 49.08 | 48.33 ± 19.26 | 122.81 ± 34.71 |
| TB | 26.29 ± 10.73 [a] | 8.64 ± 3.05 [a] | 26.52 ± 12.47 [a] |
| Single-formula-treated | | | |
| Gefitinib | 26.23 ± 10.81 [a] | 8.36 ± 1.96 [a] | 25.58 ± 11.94 [a] |
| SKOG | 70.03 ± 38.12 [acd] | 30.26 ± 12.78 [bcd] | 73.46 ± 19.50 [acd] |
| Gefitinib and SKOG | | | |
| 400 mg/kg | 98.80 ± 26.02 [cd] | 44.35 ± 10.11 [cd] | 97.56 ± 26.24 [cd] |
| 200 mg/kg | 81.62±24.83 [acd] | 37.41 ± 12.06 [cd] | 82.83 ± 22.96 [bcd] |
| 100 mg/kg | 71.75 ± 25.06 [acd] | 29.51 ± 12.75 [bcd] | 72.39 ± 12.62 [acd] |

TB = tumor-bearing; AR = Adenophorae Radix; KOG = *Kyeongokgo*; SKOG = *sasam-Kyeongokgo*. [a] $p < 0.01$ and [b] $p < 0.05$ versus intact control, assessed by the Mann–Whitney U test; [c] $p < 0.01$ versus TB control, assessed by the Mann–Whitney U test; [d] $p < 0.01$ versus gefitinib-single-formula-treated rodents, assessed by the Mann–Whitney U test.

*3.11. Histological Changes*

3.11.1. Histopathological Changes in the Tumor Mass

In the tumor transplantation control group, the tumor was densely composed of relatively well-differentiated NCI-H520 lung cancer cells. Cytoplasmic eosinophilic increase and nuclear enrichment due to apoptosis were confirmed in some cells, and mitosis was also frequently observed. In contrast, a significant increase ($p < 0.01$) in the number of apoptotic cells was confirmed in the gefitinib- and sasam-Kyeongokgo-alone groups and in all the sasam-Kyeongokgo and gefitinib combination groups compared with the tumor transplantation control group. The proportion of NCI-H520 cells was also significantly reduced ($p < 0.01$). In particular, a significant decrease ($p < 0.01$) in the tumor cell volume and an increase in the number of apoptotic cells were observed in the sasam-Kyeongokgo and gefitinib combination groups in comparison with the gefitinib-alone group (Table 6, Figure 7). In addition, the sasam-Kyeongokgo and gefitinib combination administration groups, the gefitinib single administration group, the sasam-Kyeongokgo-alone group, and the sasam-Kyeongokgo 400 mg/kg single administration group showed a significant caspase-3 and PARP immune response ($p < 0.01$) in the tumor mass. An increase in the number of cells was confirmed in comparison with that observed in the tumor transplantation control group. In particular, a significant increase ($p < 0.01$ or $p < 0.05$) in the number of PARP- and caspase-3-immunoreactive cells was confirmed in the sasam-Kyeongokgo and gefitinib combined administration groups compared with the gefitinib-alone group (Table 6, Figures 8 and 9). In the sasam-Kyeongokgo-alone and all the sasam-Kyeongokgo-co-administered groups, the tumor masses showed a significant increase ($p < 0.01$) in the number of iNOS and TNF-$\alpha$ immunoreactive cells and a decrease in the number of COX-2 immunoreactive cells in comparison with the transplant control group. In particular, the sasam-Kyeongokgo and gefitinib combination administration groups showed a significant increase ($p < 0.01$ or $p < 0.05$) in the number of iNOS and TNF-$\alpha$ immunoreactive cells and a decrease in the number of COX-2 immune-reactive cells in comparison with the gefitinib-alone administration group. On the other hand, a significant decrease ($p < 0.01$) in the number of COX-2 immunoreactive cells was confirmed in the gefitinib-alone group in comparison with the tumor transplantation control group. However, there were no significant changes in the iNOS- and TNF-$\alpha$ immune response cells (Table 6, Figures 10–12).

**Table 6.** Changes in the tumor mass histomorphometry of NCI-H520 tumor cell xenograft mice.

| Groups | Tumor Cell Volume (%/mm²) | Apoptotic Cell Percentages (%) | Immunoreactive Cell Percentages (%/tumor cells) | | | | |
|---|---|---|---|---|---|---|---|
| | | | Caspase-3 | PARP | COX-2 | iNOS | TNF-$\alpha$ |
| Control | | | | | | | |
| TB | 79.27 ± 8.85 | 7.56 ± 3.59 | 11.01 ± 2.81 | 14.52 ± 6.32 | 59.83 ± 10.29 | 12.13 ± 4.23 | 7.76 ± 1.66 |
| Single-formula-treated | | | | | | | |
| Gefitinib | 56.28 ± 7.19 [a] | 49.88 ± 8.53 [a] | 40.60 ± 11.72 [d] | 46.98 ± 10.22 [a] | 38.89 ± 5.69 [d] | 12.07 ± 5.40 | 7.28 ± 2.74 |
| SKOG | 66.24 ± 5.75 [a] | 30.82 ± 5.08 [a] | 28.77 ± 6.58 [df] | 35.84 ± 10.63 [ac] | 39.01 ± 13.59 [d] | 37.16 ± 10.24 [ab] | 37.05 ± 11.47 [de] |
| Gefitinib and SKOG | | | | | | | |
| 400 mg/kg | 29.13 ± 8.75 [ab] | 86.63 ± 7.29 [ab] | 75.42 ± 10.45 [de] | 86.29 ± 5.22 [ab] | 11.88 ± 2.99 [de] | 74.13 ± 10.12 [ab] | 62.21 ± 17.33 [de] |
| 200 mg/kg | 38.59 ± 7.28 [ab] | 75.19 ± 8.18 [ab] | 67.61 ± 12.98 [de] | 73.59 ± 10.57 [ab] | 20.69 ± 5.09 [de] | 63.79 ± 11.92 [ab] | 54.51 ± 12.48 [de] |
| 100 mg/kg | 45.75 ± 5.23 [ab] | 65.37 ± 10.60 [ab] | 56.78 ± 9.36 [df] | 62.96 ± 7.65 [ab] | 28.77 ± 7.32 [df] | 41.65 ± 11.51 [ab] | 41.97 ± 13.30 [de] |

TB = tumor-bearing; AR = Adenophorae Radix; KOG = *Kyeongokgo*; SKOG = *sasam-Kyeongokgo*; PARP = cleaved poly(ADP-ribose) polymerase; COX-2 = cyclo-oxygenase-2; iNOS = inducible nitric oxide synthase; TNF = tumor necrosis factor. [a] $p < 0.01$ versus TB control, assessed by the least significant difference (LSD) multi-comparison test; [b] $p < 0.01$ and [c] $p < 0.05$ versus gefitinib-single-formula-treated rodents assessed by the least significant difference (LSD) multi-comparison test; [d] $p < 0.01$ versus TB control assessed by the Mann–Whitney U test; [e] $p < 0.01$ and [f] $p < 0.05$ versus gefitinib-single-formula-treated rodents assessed by the Mann–Whitney U test.

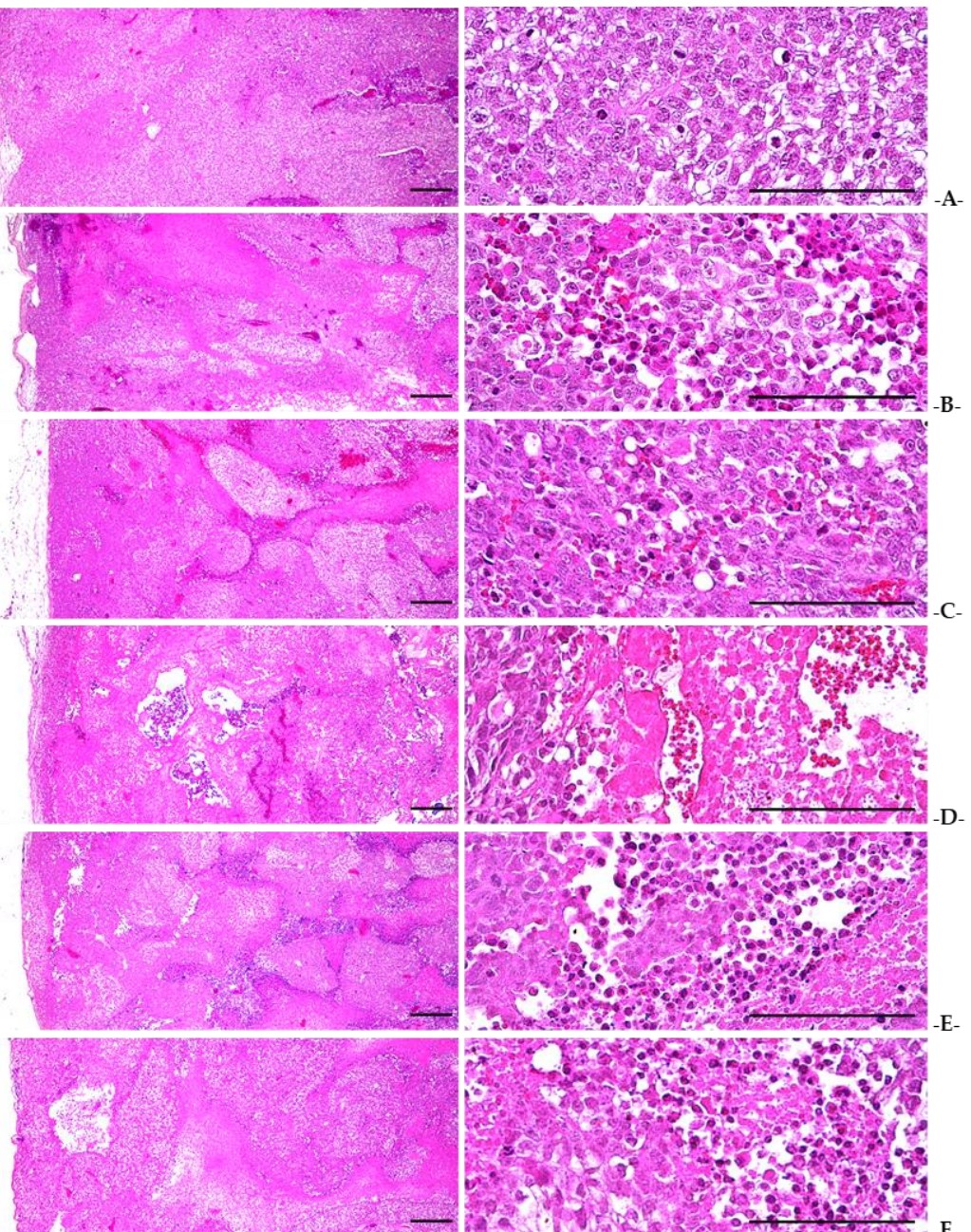

**Figure 7.** Representative histological profiles of the tumor masses taken from NCI-H520 tumor cell xenograft mice. (**A**) = Tumor-bearing control; (**B**) = 120 mg/kg-gefitinib-single-treated rodents; (**C**) = 400 mg/kg-SKOG-single-treated rodents; (**D**) = gefitinib 120 mg/kg and SKOG 400 mg/kg; (**E**) = gefitinib 120 mg/kg and SKOG 200 mg/kg; (**F**) = gefitinib 120 mg/kg and SKOG 100 mg/kg. AR = Adenophorae Radix; KOG = *Kyeongokgo*; SKOG = *sasam-Kyeongokgo*. All Hematoxylin-Eosin-stained. Scale bars = 100 μm.

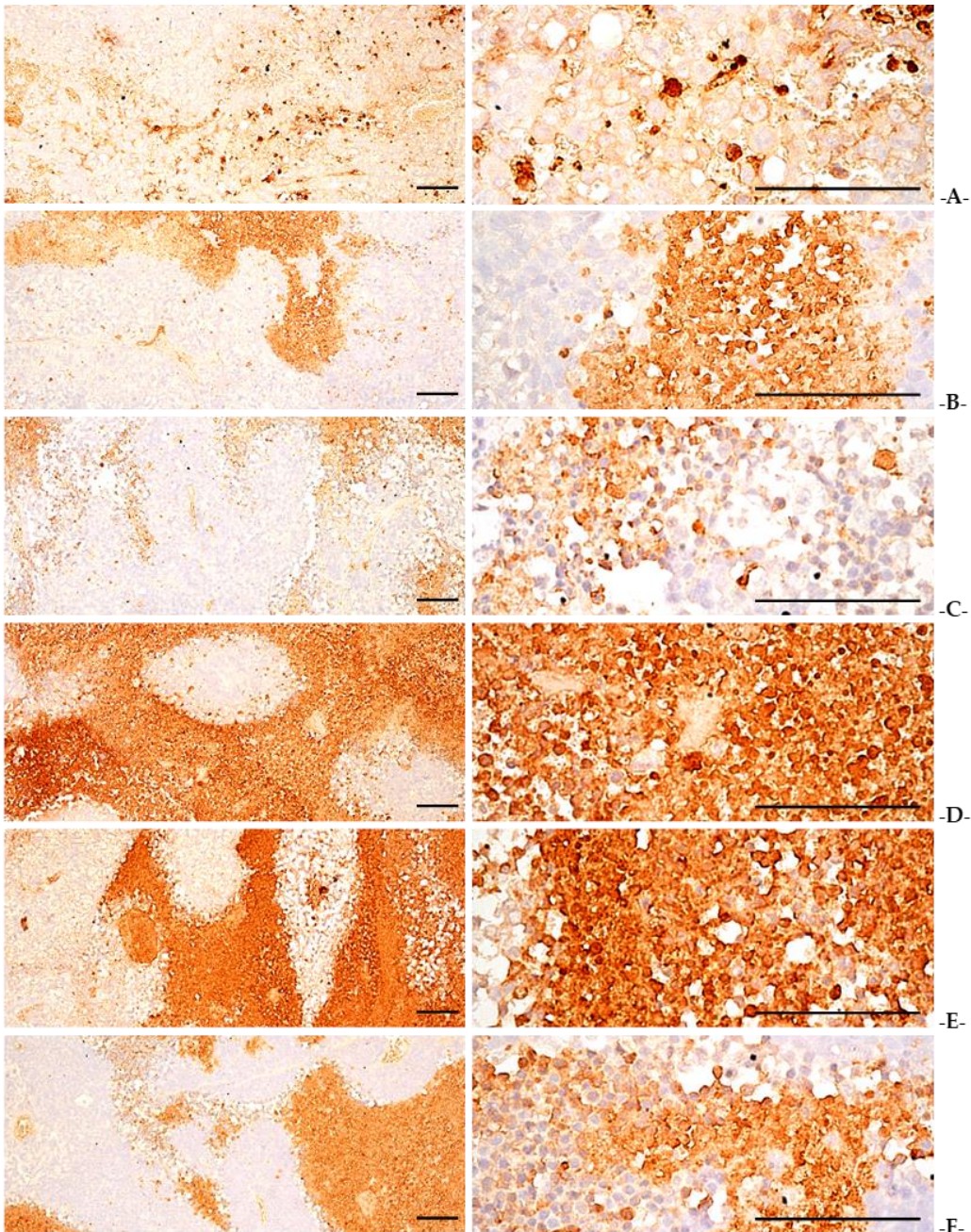

**Figure 8.** Representative caspase-3 immunoreactivities of the tumor masses taken from NCI-H520 tumor cell xenograft mice. (**A**) = Tumor-bearing control; (**B**) = 120 mg/kg-gefitinib-single-treated rodents; (**C**) = 400 mg/kg-SKOG-single-treated rodents; (**D**) = gefitinib 120 mg/kg and SKOG 400 mg/kg; (**E**) = gefitinib 120 mg/kg and SKOG 200 mg/kg; (**F**) = gefitinib 120 mg/kg and SKOG 100 mg/kg. AR = Adenophorae Radix; KOG = *Kyeongokgo*; SKOG = *sasam-Kyeongokgo*. All avidin-biotin complex (ABC) methods. Scale bars = 100 μm.

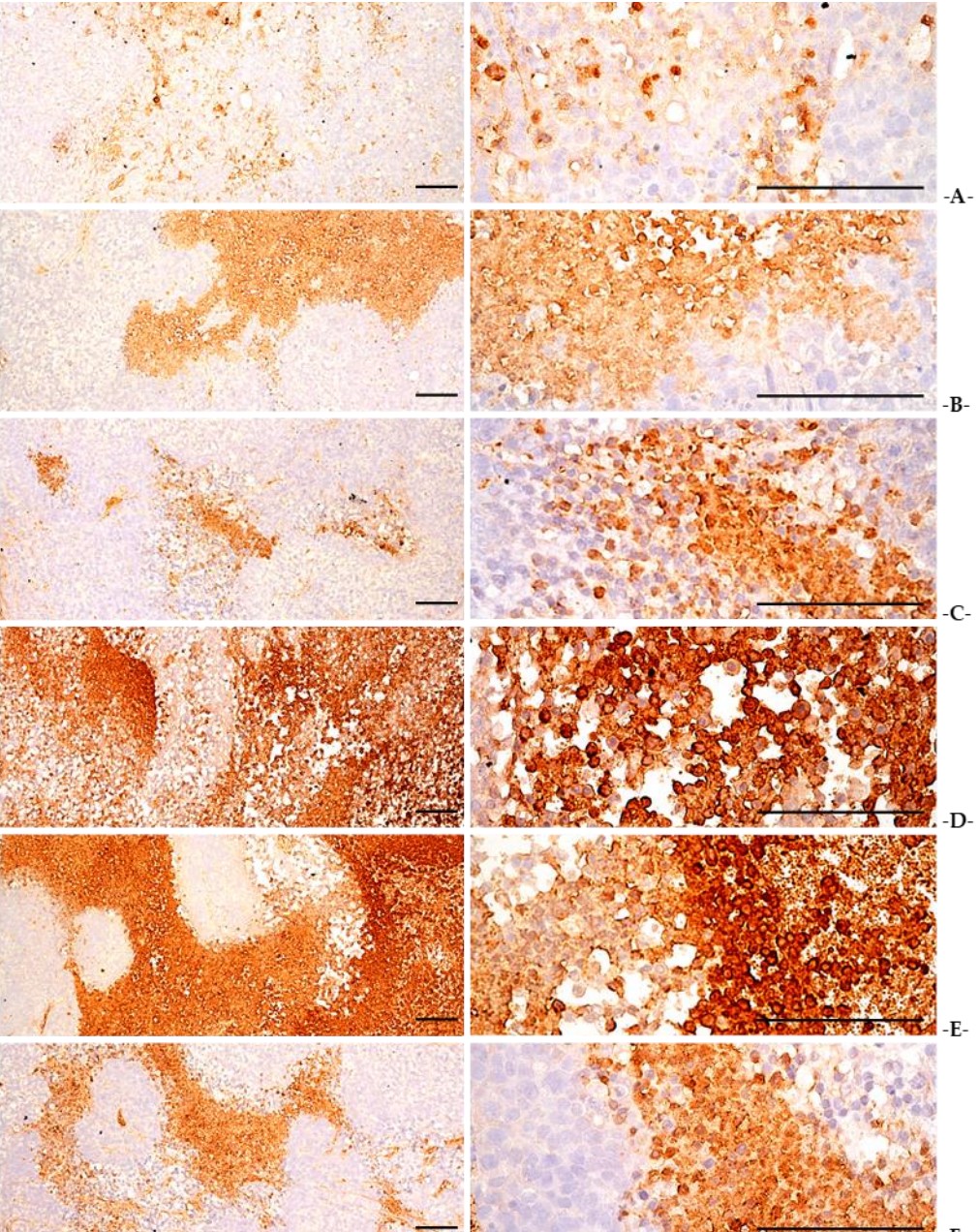

**Figure 9.** Representative PARP immunoreactivities of the tumor masses taken from NCI-H520 tumor cell xenograft mice. (**A**) = Tumor-bearing control; (**B**) = 120 mg/kg-gefitinib-single-treated rodents; (**C**) = 400 mg/kg-SKOG-single-treated rodents; (**D**) = gefitinib 120 mg/kg and SKOG 400 mg/kg; (**E**) = gefitinib 120 mg/kg and SKOG 200 mg/kg; (**F**) = gefitinib 120 mg/kg and SKOG 100 mg/kg. AR = Adenophorae Radix; KOG = *Kyeongokgo*; SKOG = *sasam-Kyeongokgo*. PARP = cleaved poly(ADP-ribose) polymerase. All avidin-biotin complex (ABC) methods. Scale bars = 100 μm.

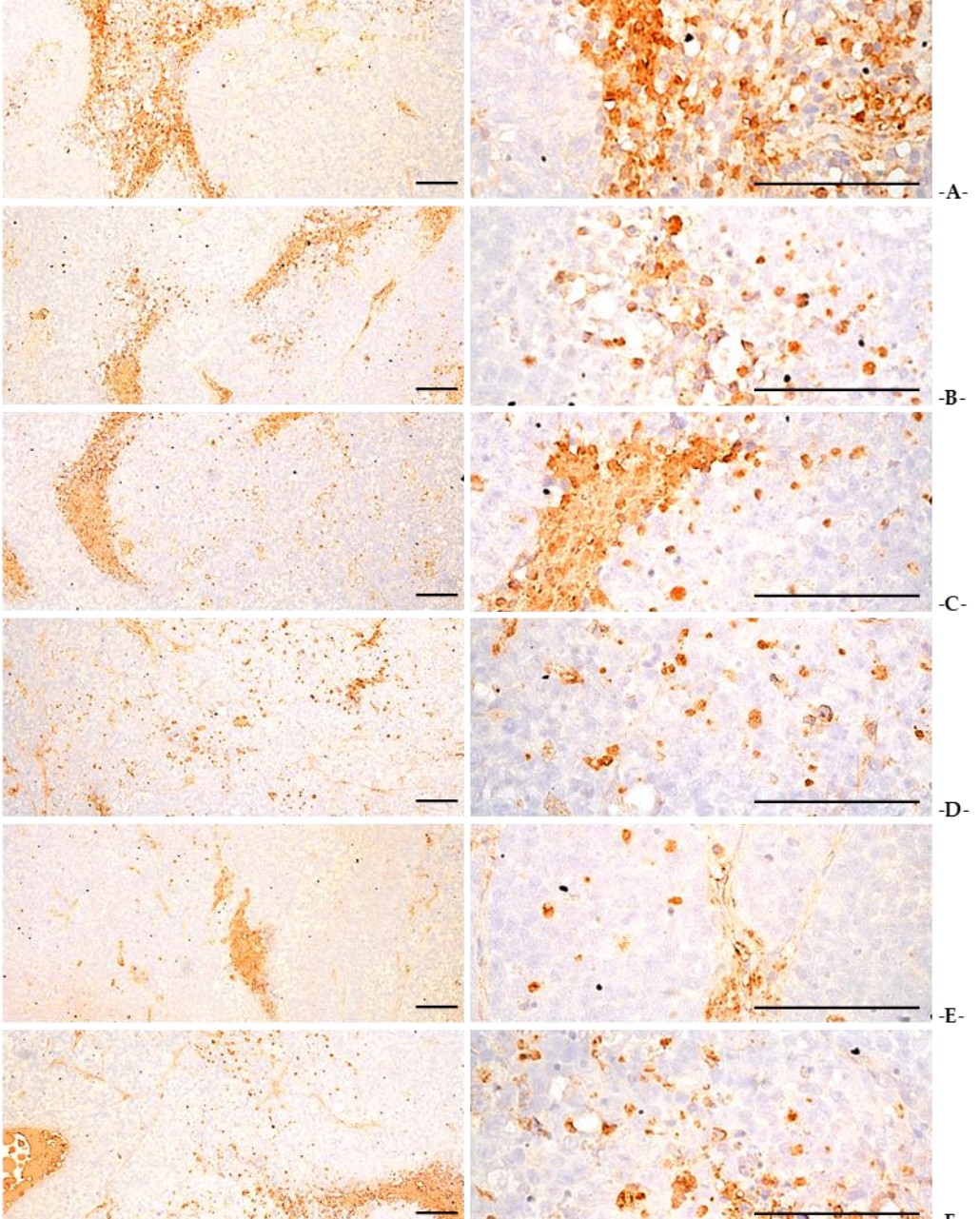

**Figure 10.** Representative COX-2-immunoreactivities of the tumor masses taken from NCI-H520 tumor cell xenograft mice. (**A**) = Tumor-bearing control; (**B**) = 120 mg/kg-gefitinib-single-treated rodents; (**C**) = 400 mg/kg-SKOG-single-treated rodents; (**D**) = gefitinib 120 mg/kg and SKOG 400 mg/kg; (**E**) = gefitinib 120 mg/kg and SKOG 200 mg/kg; (**F**) = gefitinib 120 mg/kg and SKOG 100 mg/kg. AR = Adenophorae Radix; KOG = *Kyeongokgo*; SKOG = *sasam-Kyeongokgo*. COX-2 = cyclo-oxygenase-2. All avidin-biotin complex (ABC) methods. Scale bars = 100 μm.

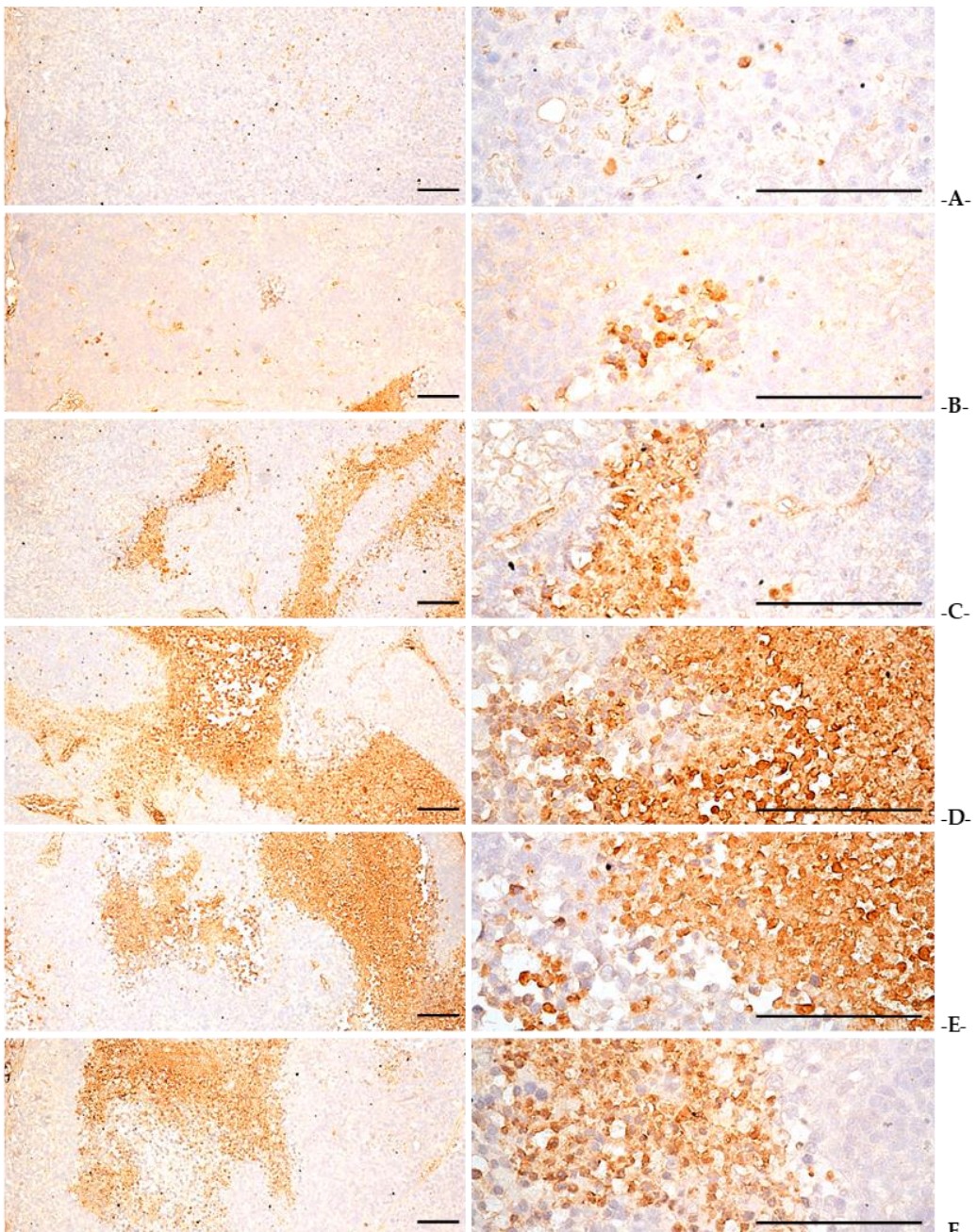

**Figure 11.** Representative iNOS immunoreactivities of the tumor masses taken from NCI-H520 tumor cell xenograft mice. (**A**) = Tumor-bearing control; (**B**) = 120 mg/kg-gefitinib-single-treated rodents; (**C**) = 400 mg/kg-SKOG-single-treated rodents; (**D**) = gefitinib 120 mg/kg and SKOG 400 mg/kg; (**E**) = gefitinib 120 mg/kg and SKOG 200 mg/kg; (**F**) = gefitinib 120 mg/kg and SKOG 100 mg/kg. AR = Adenophorae Radix; KOG = *Kyeongokgo*; SKOG = *sasam-Kyeongokgo*. iNOS = inducible nitric oxide synthase. All avidin-biotin complex (ABC) methods. Scale bars = 100 μm.

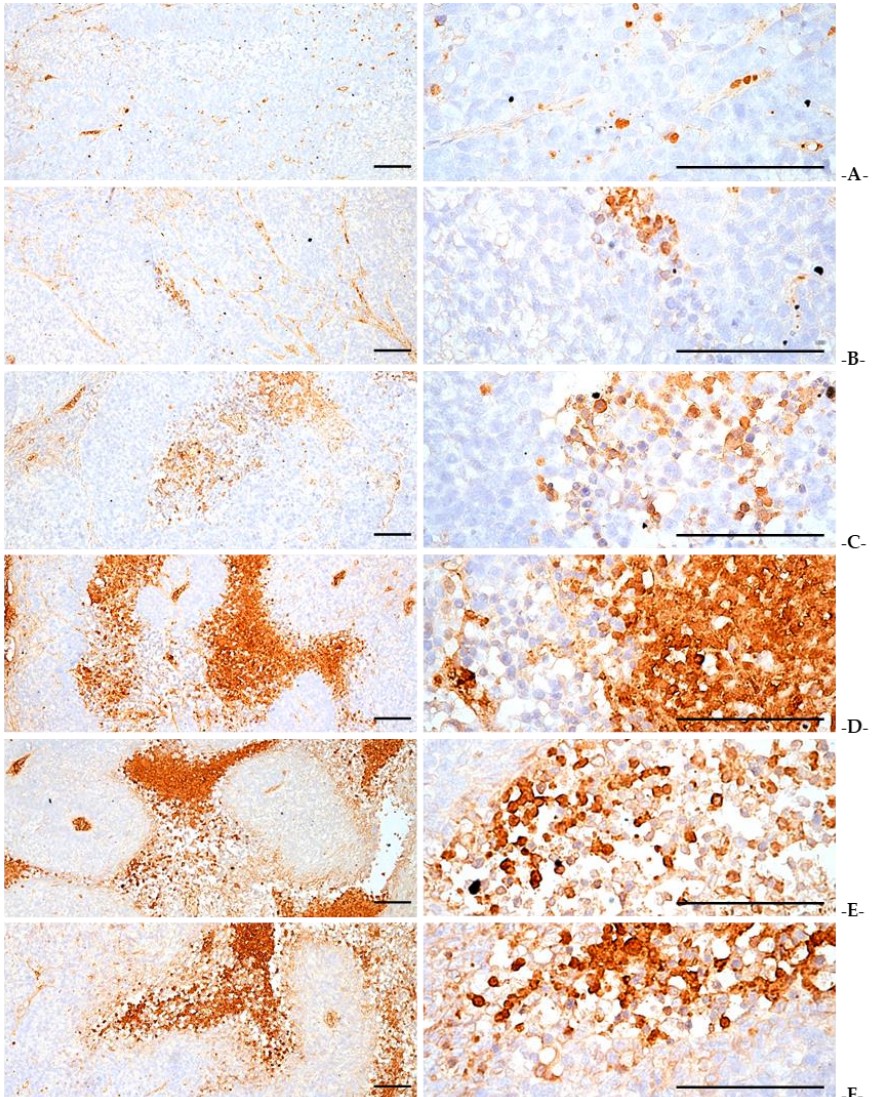

**Figure 12.** Representative TNF-α immunoreactivities of the tumor masses taken from NCI-H520 tumor cell xenograft mice. (**A**) = Tumor-bearing control; (**B**) = 120 mg/kg-gefitinib-single-treated rodents; (**C**) = 400 mg/kg-SKOG-single-treated rodents; (**D**) = gefitinib 120 mg/kg and SKOG 400 mg/kg; (**E**) = gefitinib 120 mg/kg and SKOG 200 mg/kg; (**F**) = gefitinib 120 mg/kg and SKOG 100 mg/kg. AR = Adenophorae Radix; KOG = *Kyeongokgo*; SKOG = *sasam-Kyeongokgo*. TNF = tumor necrosis factor. All avidin-biotin complex (ABC) methods. Scale bars = 100 μm.

### 3.11.2. Histopathological Changes in the Spleen

In the tumor transplantation control group, atrophy characterized by a significant decrease in the number of lymphocytes in the white medullary part of the spleen was confirmed, and significant reductions ($p < 0.01$) in the spleen thickness, white medullary diameter, and number were confirmed in comparison with the intact control group. However, significant increments ($p < 0.01$) in the spleen thickness, white medullary diameter, and number were histopathologically confirmed in the sasam-Kyeongokgo-alone and all the combination groups in comparison with the tumor transplantation control group. In particular, significant increments ($p < 0.01$) in the spleen thickness, white medullary diameter, and number were observed in all the sasam-Kyeongokgo and gefitinib combination groups in comparison with the gefitinib-alone group. In contrast, changes in the spleen thickness, white medullary diameter, and number similar to those in the tumor transplantation control group were observed in the gefitinib-alone group (Table 7, Figure 13).

**Table 7.** Changes in the spleen histomorphometry of NCI-H520 tumor cell xenograft mice.

| Groups | Total Thickness (mm/Central Regions) | White Pulp Numbers (/mm²) | White Pulp Diameters (μm/White Pulp) |
|---|---|---|---|
| Controls | | | |
| Intact | 1749.75 ± 190.38 | 14.75 ± 2.92 | 645.75 ± 127.16 |
| TB | 989.25 ± 135.20 [a] | 4.88 ± 1.13 [a] | 188.75 ± 69.98 [e] |
| Single-formula-treated | | | |
| Gefitinib | 993.38 ± 131.85 [a] | 4.63 ± 1.85 [a] | 187.63 ± 73.54 [e] |
| SKOG | 1281.50 ± 123.37 [acd] | 8.25 ± 1.67 [acd] | 323.00 ± 66.08 [efg] |
| Gefitinib and SKOG co-administered within 5 min | | | |
| 400 mg/kg | 1489.75 ± 136.36 [acd] | 13.88 ± 2.64 [cd] | 449.50 ± 38.91 [efg] |
| 200 mg/kg | 1340.38 ± 77940 [acd] | 12.75 ± 1.28 [bcd] | 404.75 ± 50.76 [efg] |
| 100 mg/kg | 1276.13 ± 201.59 [acd] | 8.63 ± 1.30 [acd] | 352.50 ± 75.45 [efg] |

TB = tumor-bearing; AR = Adenophorae Radix; KOG = *Kyeongokgo*; SKOG = *sasam-Kyeongokgo*. [a] $p < 0.01$ and [b] $p < 0.05$ versus intact control, assessed by the least significant difference (LSD) multi-comparison test; [c] $p < 0.01$ versus TB control, assessed by the least significant difference (LSD) multi-comparison test; [d] $p < 0.01$ versus gefitinib-single-formula-treated rodents, assessed by the least significant difference (LSD) multi-comparison test; [e] $p < 0.01$ versus intact control, assessed by the Mann–Whitney U test; [f] $p < 0.01$ versus TB control, assessed by the Mann–Whitney U test; [g] $p < 0.01$ versus gefitinib-single-formula-treated rodents, assessed by the Mann–Whitney U test.

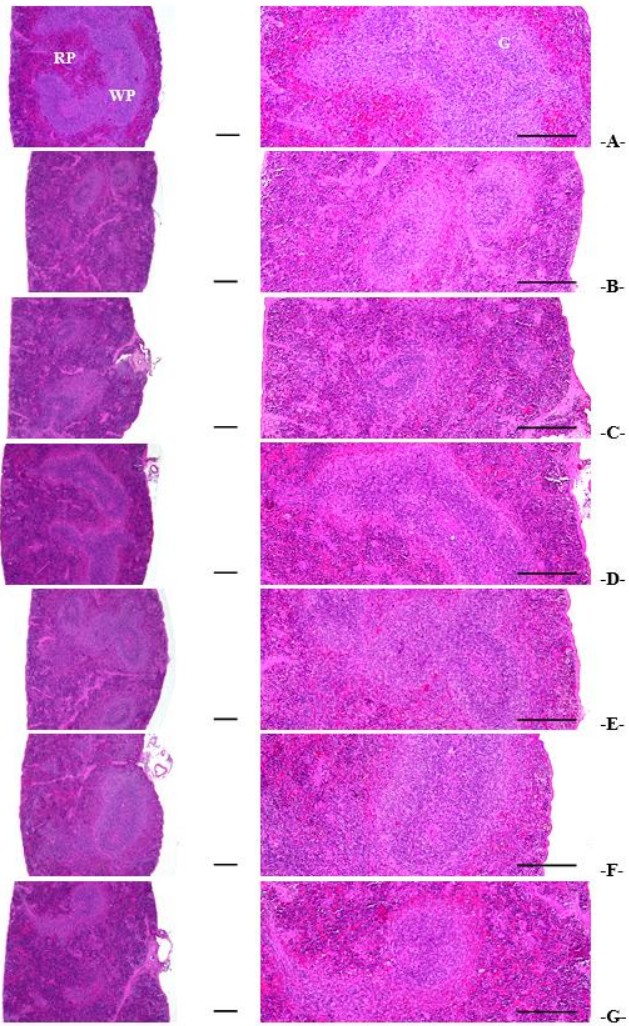

**Figure 13.** Representative histological profiles of the spleens taken from intact or NCI-H520 tumor cell xenograft mice. (**A**) = Intact control; (**B**) = tumor-bearing control; (**C**) = gefitinib 120 mg/kg; (**D**) SKOG 400 mg/kg; (**E**) = gefitinib 120 mg/kg and SKOG 400 mg/kg; (**F**) = gefitinib 120 mg/kg and SKOG 200 mg/kg; (**G**) = gefitinib 120 mg/kg and SKOG 100 mg/kg. AR = Adenophorae Radix; KOG = *Kyeongokgo*; SKOG = *sasam-Kyeongokgo*. WP = white pulp; RP = red pulps; G = secondary follicle. All Hematoxylin-Eosin-stained. Scale bars = 200 μm.

### 3.11.3. Histopathological Changes in the Submandibular Lymph Nodes

In the tumor transplantation control group, atrophy was confirmed due to a significant decrease in the number of lymphocytes in the lymph node cortex, and a significant decrease ($p < 0.01$) in the total submandibular lymph nodes and cortical thickness and the number of intracortical follicles were confirmed in comparison with the intact control group. However, in the group that received sasam-Kyeongokgo alone or in combination, significant increases ($p < 0.01$) in the total lymph nodes, cortical thickness, and intracortical follicle number were histopathologically confirmed compared with the tumor transplantation control group. In particular, in the group administered sasam-Kyeongokgo and gefitinib, significant increases ($p < 0.01$) in the total lymph nodes, cortical thickness, and intracortical follicle number were observed in comparison with the group administered gefitinib alone. In contrast, no significant changes in the total lymph nodes, cortical thickness, or intracortical follicle number were confirmed in the gefitinib-alone group compared with the tumor transplantation control group (Table 8, Figure 14).

**Table 8.** Changes in the submandibular lymph node histomorphometry of NCI-H520 tumor cell xenograft mice.

| Groups | Total Thickness (μm/Central Regions) | Cortex Lymphoid Cell Follicle Numbers (/mm²) | Cortex Thickness (μm/Lymph Node) |
|---|---|---|---|
| Controls | | | |
| Intact | 1127.63 ± 131.38 | 16.00 ± 3.93 | 540.50 ± 113.31 |
| TB | 514.63 ± 101.14 [a] | 5.75 ± 1.28 [d] | 224.38 ± 41.00 [d] |
| Single-formula-treated | | | |
| Gefitinib | 501.50 ± 106.64 [a] | 5.38 ± 2.39 [d] | 215.75 ± 43.93 [d] |
| SKOG | 744.00 ± 111.81 [abc] | 8.63 ± 1.30 [dfg] | 306.88 ± 25.76 [dfg] |
| Gefitinib and SKOG co-administered within 5 min | | | |
| 400 mg/kg | 843.50 ± 113.09 [abc] | 13.00 ± 2.73 [fg] | 423.25 ± 46.48 [efg] |
| 200 mg/kg | 868.63 ± 113.53 [abc] | 10.50 ± 1.93 [dfg] | 380.75 ± 43.65 [dfg] |
| 100 mg/kg | 727.51 ± 103.85 [abc] | 8.88 ± 1.25 [dfg] | 328.38 ± 49.75 [dfg] |

TB = tumor-bearing; AR = Adenophorae Radix; KOG = *Kyeongokgo*; SKOG = *sasam-Kyeongokgo*. [a] $p < 0.01$ versus intact control, assessed by the least significant difference (LSD) multi-comparison test; [b] $p < 0.01$ versus TB control, assessed by the least significant difference (LSD) multi-comparison test; [c] $p < 0.01$ versus gefitinib-single-formula-treated rodents, assessed by the least significant difference (LSD) multi-comparison test; [d] $p < 0.01$ and [e] $p < 0.05$ versus intact control, assessed by the Mann–Whitney U test; [f] $p < 0.01$ versus TB control, assessed by the Mann–Whitney U test; [g] $p < 0.01$ versus gefitinib-single-formula-treated rodents, assessed by the Mann–Whitney U test.

### 3.11.4. Histopathological Changes in Periovarian Fat

In the tumor transplantation control group, atrophy characterized by a significant decrease in the size of the white adipocytes was confirmed, and significant decreases ($p < 0.01$) in the accumulated fat thickness and mean diameter of the white adipocytes were confirmed in comparison with the intact control group. Meanwhile, significant increments ($p < 0.01$) in the accumulated fat thickness and average diameter of the white adipocytes were confirmed histopathologically in the sasam-Kyeongokgo alone and sasam-Kyeongokgo with gefitinib groups, respectively, in comparison with the tumor transplantation control group. In particular, significant increments ($p < 0.01$) in the accumulated fat thickness and average diameter of the white adipocytes were observed in the group administered sasam-Kyeongokgo and gefitinib in comparison with the group administered gefitinib alone. On the other hand, in the gefitinib-alone group, we observed changes in the thickness of the periovarian adipose tissue and the mean diameter of the white adipocytes similar to those observed in the tumor transplantation control group (Table 9, Figure 15).

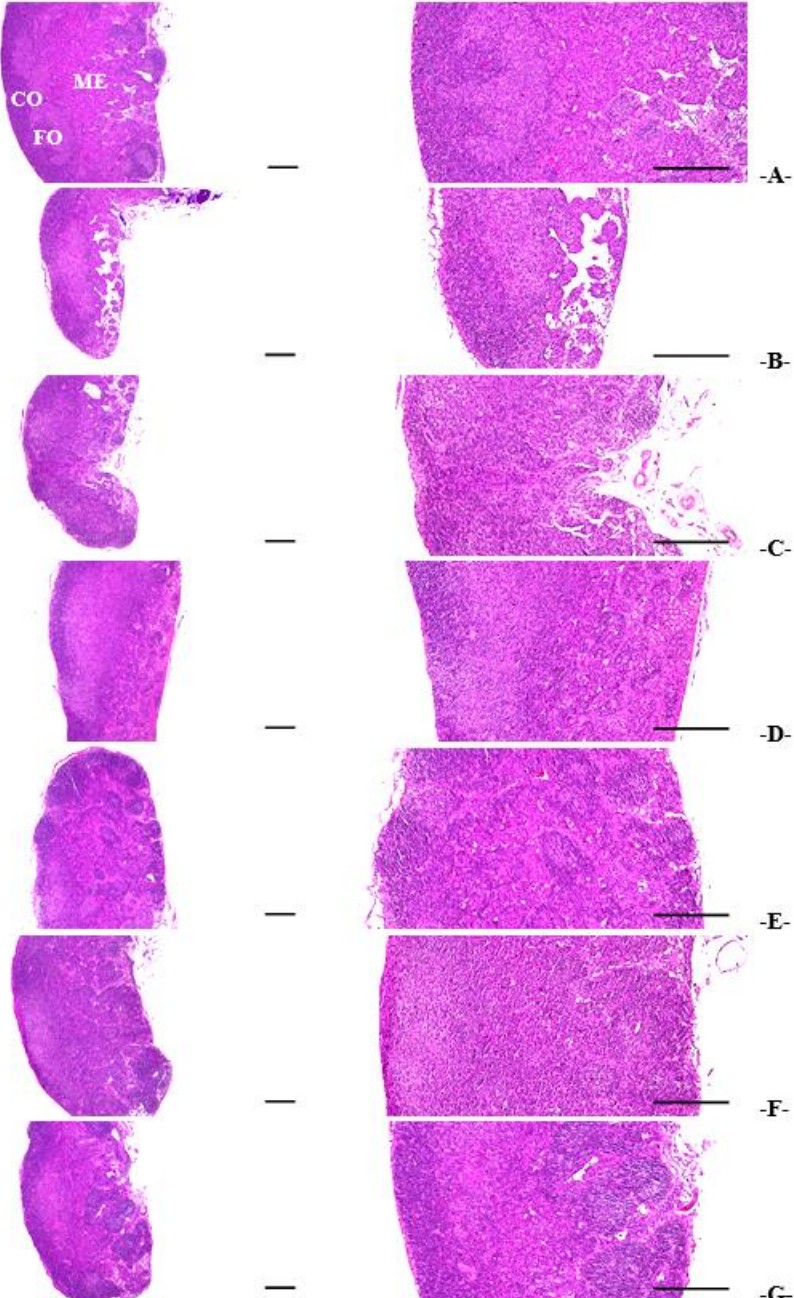

**Figure 14.** Representative histological profiles of the submandibular lymph nodes taken from intact or NCI-H520 tumor cell xenograft rodents. (**A**) = Intact control; (**B**) = tumor-bearing control; (**C**) = gefitinib 120 mg/kg; (**D**) = SKOG 400 mg/kg; (**E**) = gefitinib 120 mg/kg and SKOG 400 mg/kg; (**F**) = gefitinib 120 mg/kg and SKOG 200 mg/kg; (**G**) = gefitinib 120 mg/kg and SKOG 100 mg/kg. AR = Adenophorae Radix; KOG = *Kyeongokgo*; SKOG = *sasam-Kyeongokgo*. CO = cortex; ME = medullar; FO = follicle. All Hematoxylin-Eosin-stained. Scale bars = 400 μm.

**Table 9.** Changes in the periovarian fat pad histomorphometry of NCI-H520 tumor cell xenograft mice.

| Groups | Total Thickness (mm/Central Regions) | White Adipocyte Diameters (μm) |
|---|---|---|
| Controls | | |
| Intact | 1752.13 ± 201.31 | 52.11 ± 5.21 |
| TB | 409.25 ± 103.05 [a] | 13.20 ± 2.61 [d] |

**Table 9.** *Cont.*

| Groups | Total Thickness (mm/Central Regions) | White Adipocyte Diameters (μm) |
|---|---|---|
| Single-formula-treated | | |
| Gefitinib | 375.75 ± 91.45 [a] | 12.36 ± 1.95 [d] |
| SKOG | 650.00 ± 121.29 [abc] | 21.85 ± 3.14 [def] |
| Gefitinib and SKOG co-administered within 5 min | | |
| 400 mg/kg | 1453.00 ± 192.34 [abc] | 37.95 ± 10.89 [def] |
| 200 mg/kg | 1096.00 ± 128.90 [abc] | 32.51 ± 11.42 [def] |
| 100 mg/kg | 965.00 ± 192.82 [abc] | 25.10 ± 6.62 [def] |

TB = tumor-bearing; AR = Adenophorae Radix; KOG = *Kyeonggokgo*; SKOG = *sasam-Kyeongokgo*. [a] $p < 0.01$ versus intact control, assessed by the least significant difference (LSD) multi-comparison test; [b] $p < 0.01$ versus TB control assessed by the least significant difference (LSD) multi-comparison test; [c] $p < 0.01$ versus gefitinib-single-formula-treated rodents, assessed by the least significant difference (LSD) multi-comparison test; [d] $p < 0.01$ versus intact control, assessed by the Mann–Whitney U test; [e] $p < 0.01$ versus TB control, assessed by the Mann–Whitney U test; [f] $p < 0.01$ versus gefitinib-single-formula-treated rodents, assessed by the Mann–Whitney U test.

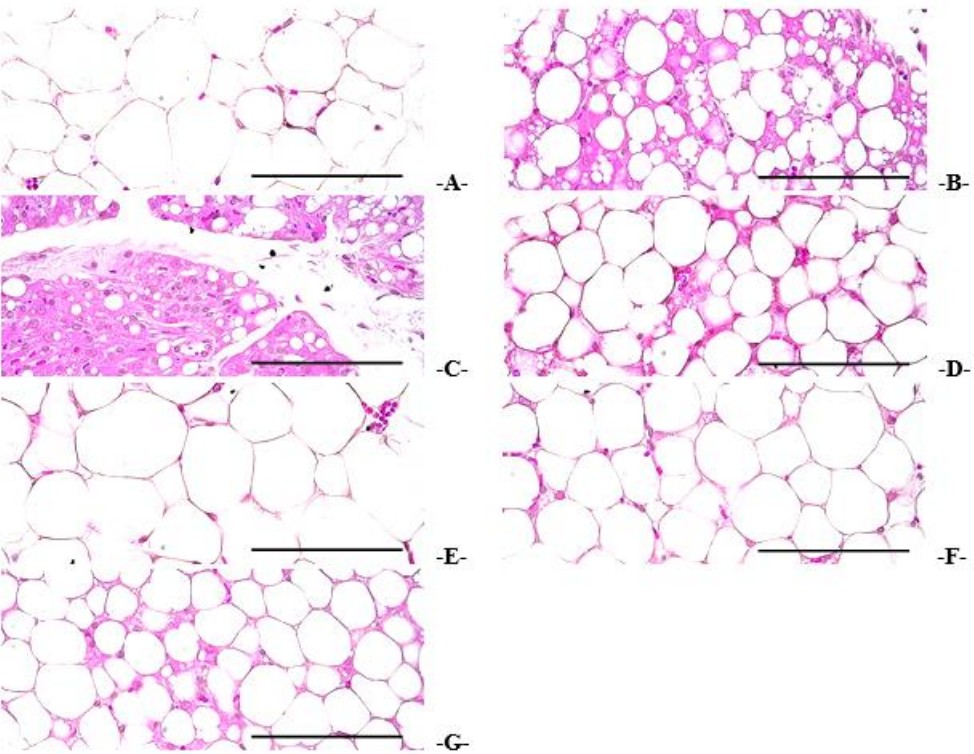

**Figure 15.** Representative histological profiles of the periovarian fat pads taken from intact or NCI-H520 tumor cell xenograft mice. (**A**) = Intact control; (**B**) = tumor-bearing control; (**C**) = gefitinib 120 mg/kg; (**D**) = SKOG 400 mg/kg; (**E**) = Gefitinib 120 mg/kg and SKOG 400 mg/kg; (**F**) = Gefitinib 120 mg/kg and SKOG 200 mg/kg; (**G**) = gefitinib 120 mg/kg and SKOG 100 mg/kg. AR = Adenophorae Radix; KOG = *Kyeongokgo*; SKOG = *sasam-Kyeongokgo*. All Hematoxylin-Eosin-stained. Scale bars = 100 μm.

## 4. Discussion

Gefitinib is a representative EGFR inhibitor and oral anticancer drug that is frequently used to treat various malignancies, including lung and breast cancers, and is known to mainly inhibit the EGFR tyrosine kinase domain. As a target-oriented anticancer agent, gefitinib is also known to have much lower toxicity than conventional cytotoxic anticancer agents [31,32]. However, the drug has been reported to cause various unintended side effects, including skin rashes, diarrhea, nausea, vomiting, anorexia, gastritis, dehydration, paronychia, liver toxicity, asthenia, conjunctivitis, blepharitis, interstitial lung disease,

corneal erosion, and eyelash loss [1,2]. Hypersensitivity to gefitinib has also been reported [33,34]. Other studies have suggested that the drug causes liver toxicity due to the increase in lipid peroxidation due to metabolites formed in the liver and damage to the antioxidant defense system [35]. With the recent emergence of resistant malignant tumor cells due to mutations of EGFR [36,37], attempts have been made to solve the toxicity and tolerance problems associated with gefitinib through the co-administration of natural products and drugs, including various antioxidants [38,39].

Sasam-Kyeongokgo is a mixed prescription of Kyeongokgo, a typical complex prescription that has traditionally been taken for health promotion in Korea [10,11], and sasam, which has been utilized for the treatment of various respiratory diseases [3,5]. In this study, as part of the effort to develop new effective anticancer therapies for lung cancer patients, the effects of sasam-Kyeongokgo on the anticancer activity of gefitinib were investigated in the NCI-H520 cell line, a representative squamous cell carcinoma non-small-cell lung cancer cell line showing gefitinib resistance [40,41].

Cytotoxicity evaluation involves the assessment of a candidate substance's toxicity to cells. An anticancer drug should have low cytotoxicity to normal cells and should selectively exhibit cytotoxicity to malignant tumor cells [42]. In this experiment, the $IC_{50}$ values of sasam-Kyeongokgo and gefitinib for NCI-H520 cells were calculated as $17.21 \pm 7.28$ mg/mL and $4.56 \pm 1.46$ μM ($2.00 \pm 0.64$ μg/mL), respectively. Thus, both drugs exhibited relatively low cytotoxicity to the cells. In general, gefitinib is known to have an $IC_{50}$ of approximately 0.1 μM for sensitive EGFR-expressing tumor cell lines. The NCI-H520 cells used in this experiment also exhibited resistance to gefitinib, similar to the finding of a previous report [41].

Nude mice are typical athymic animals and are the most commonly used experimental animals for the development of anticancer drugs, since they can receive transplants of allogeneic tumor cells as well as human-derived tumor cells [43]. In a tumor-transplanted nude mouse model, the anticancer effect is mainly characterized by the growth inhibition of the transplanted mass [44,45]. The results of this experiment confirmed that the sasam-Kyeongokgo and gefitinib combination administration groups showed significantly superior reductions in the tumor volume and tumor weight in comparison with the gefitinib-alone group. In the cytotoxicity test for the NCI-H520 cells, sasam-Kyeongokgo showed relatively low cytotoxicity, but increases in the number of apoptotic cells in the tumor mass and immune activity were confirmed through various test results. On the basis of these findings, the synergistic anticancer effect of the combination of gefitinib and sasam-Kyeongokgo is considered to be due to immune activity rather than direct cytotoxicity to tumor cells.

Significant immunosuppression is known to be induced after tumor transplantation, and this immunosuppression is mainly related to the T lymphocytes [46]. Various studies have been conducted to identify measures that can be implemented to strengthen the body's defense against tumors by stimulating an individual's immune cells and inducing cytokine production. The development of anticancer drugs through immune activation has attracted attention in this regard [47,48]. Moreover, antioxidants are known to exhibit effective anticancer effects in relation to immune activity [49,50]. This experiment also showed significant immunosuppression caused by NCI-H520 tumor cell transplantation. Along with a decrease in the weight of the immune organs, significant atrophy was confirmed due to the decrease in the number of lymphocytes in the spleen and lymph nodes. In contrast, gefitinib alone had no significant effect on tumor-transplant-related immunosuppression. On the other hand, significant immunological activity was confirmed in the group administered sasam-Kyeongokgo alone and those that received the three doses of the sasam-Kyeongokgo and gefitinib combination. In particular, a significant immunoactive effect was confirmed in comparison with the gefitinib-alone group, and it was in good agreement with these immunoactive effects. A decrease in the tumor weight and volume and an increase in apoptosis were confirmed. Therefore, it is considered that the increase in

the anticancer effect of gefitinib by the combined administration of sasam-Kyeongokgo is related to immune activity.

TNF-$\alpha$ is a representative type of cytokine produced in various cell types, including splenocytes, and is known to play an important role in the differentiation of T lymphocytes [51]. TNF-$\alpha$ generally activates cellular immunity and increases the function of IL-2, which promotes antibody production [52]. IL-1 is another type of cytokine secreted by various cells, such as lymphocytes, dendritic cells, macrophages, endothelial cells, fibroblasts, and keratinocytes. There are two types of IL-1, namely IL-1$\beta$, secreted by cells, and IL-1$\alpha$, which is a membrane-attached type. They play important roles in the immune response [53]. IL-10 is a typical immunosuppressive cytokine secreted by Th2 cells, specific B lymphocytes, and activated macrophages and is known to inhibit the functions of activated macrophages [52]. IFN-$\gamma$ is secreted by CD8+ T lymphocytes, NK cells, and Th1 cells. It affects the functions of T and B cells and enhances the functions of NK cells and macrophages [52].

Similar to the findings of previous reports [46,47], the transplantation of NCI-H520 cells, which are human lung cancer cells, resulted in a decrease in the splenic contents of the immunoactive cytokines IL-1$\beta$ and TNF-$\alpha$ and the blood content of IFN-$\gamma$. In addition, a decrease in the spleen levels of IL-10, an immunosuppressive cytokine, caused by the T lymphocytes and immunosuppression was also confirmed. In contrast, the decreases in the splenic contents of TNF-$\alpha$, IL-1$\beta$, and IL-10 and the blood content of IFN-$\gamma$ were significantly inhibited in all the sasam-Kyeongokgo administration groups. In particular, significant increments in the levels of TNF-$\alpha$, IL-10, and IL-1$\beta$ in the spleen and IFN-$\gamma$ in all three sasam-Kyeongokgo combined administration groups were confirmed in comparison with the gefitinib-alone group.

As part of tumor-related immunosuppression, the functions of immune cells such as NK cells and macrophages are suppressed during tumor development. The activity of these immune cells has received attention in the development of newer concepts for anticancer drugs [48,49]. In this experiment, a decrease in the activity of the splenocytes and abdominal macrophages was confirmed in the tumor transplantation control group, and the gefitinib-alone group exhibited a change in NK cell activity similar to that observed in the tumor transplantation control group. However, dose-dependent increases in the abdominal and spleen NK cell activity were confirmed in all the sasam-Kyeongokgo administration groups, and significant dose-dependent increments in NK cell activity were confirmed in the group that received both sasam-Kyeongokgo and gefitinib in comparison with the gefitinib-alone group.

Caspase-3 and PARP are representative apoptosis markers [54], and an increase in caspase-3 and PARP immunoreactivity in the tumor mass indicates the apoptosis of tumor cells [55]. The results of this experiment also confirmed an increase in caspase-3 and PARP immunoreactivity in relation to the administration of gefitinib or sasam-Kyeongokgo. In particular, significant increases in intratumoral PARP and caspase-3 immunoreactivity were observed in the group administered a combination of sasam-Kyeongokgo and gefitinib, respectively, in comparison with the group administered gefitinib alone. The anticancer effect of gefitinib was significantly increased by the combined administration of sasam-Kyeongokgo at 100 mg/kg or more.

The inhibition of the immunoreactivity of COX-2 [56], which plays an important role in the synthesis of prostaglandins, a representative inflammatory mediator, and is involved in angiogenesis in tumors, was confirmed in all the sasam-Kyeongokgo groups. In particular, a significant decrease was observed in the group administered sasam-Kyeongokgo and gefitinib in comparison with the group administered gefitinib alone. A significant decrease in the number of COX-2-immunoreactive cells in the tumor tissue was also confirmed in the gefitinib-alone group in comparison with the tumor transplantation control group. In general, increased iNOS activity is linked to endotoxins, IL-1$\beta$, IFN-$\gamma$, and TNF-$\alpha$, causing shock and excessive inflammation [57,58], and is known to exacerbate angiogenesis in tumors [59]. However, iNOS secreted from immune-active cells, such as

the macrophages, has been shown to induce the apoptosis of tumor cells, resulting in tumor growth inhibition [60]. In this study, a significant increase in iNOS immunoreactivity in the NCI-H520 cell transplantation mass was confirmed in all the groups administered the test substance, except for the group administered gefitinib alone. In particular, a significant increase in intratumoral iNOS immunoreactivity was observed in the group administered the combinations of sasam-Kyeongokgo and gefitinib compared to the group administered gefitinib alone. This increase in iNOS immune reactivity is considered to be a result of immune activity following the administration of sasam-Kyeongokgo. In addition, an increase in the immunoreactivity of TNF-$\alpha$ [61], which promotes tumor necrosis in a significant tumor mass, was also confirmed in the group administered sasam-Kyeongokgo and gefitinib compared to the group treated with gefitinib alone. In contrast, in the group administered gefitinib alone, there were no significant changes in the number of iNOS and TNF-$\alpha$ immune response cells in the tumor mass compared to the tumor transplantation control group.

Cachexia, which is a very important side effect of tumors, is the most important factor that lowers the quality of life of patients with malignant tumors. Cachexia is known to cause various chronic complications, including malnutrition, dehydration, and weight loss [62,63]. To date, various studies [22,64] have shown that IL-6 produced and secreted by tumor cells is the cause of tumor-related cachexia, and a significant increase in the concentration of IL-6 in the blood occurs even in patients with actual tumors [62]. The results of this experiment confirmed that after tumor transplantation, a significant increase in the IL-6 content in the blood was accompanied by a decrease in BW, atrophy, and a decrease in the accumulated adipose tissue. However, this tumor-related cachexia was significantly suppressed in all the sasam-Kyeongokgo administration groups. In particular, the group administered sasam-Kyeongokgo and gefitinib showed a significant increase in BW, reduced serum IL-6 content, and increases in the amount of accumulated adipose tissue and the diameter of the adipocytes compared to the group administered gefitinib alone. In contrast, the group administered gefitinib alone showed no significant effect of NCI-H520 cell transplantation on tumor-related cachexia. This study has several limitations. Since NCI-H520 cells have a low expression of EGFR and related resistance to gefitinib [65], the synergistic effects of SKOG and gefitinib on the other NSCLC cell lines showed an acquired resistance to gefitinib, along with the combination index (CI index). More detailed and abundant in vitro and in vivo experiments, including efficacy tests of the major component of the SKOG, are required in the future.

## 5. Conclusions

The combination of sasam-Kyeongokgo with gefitinib was shown to have a substantial synergistic anticancer effect in comparison with treatment using gefitinib alone. Thus, the adverse hepatotoxic effects of gefitinib can be significantly reduced by reducing the gefitinib dose and co-administering the same dose of gefitinib with sasam-Kyeongokgo.

**Supplementary Materials:** The following supporting information can be downloaded at: https://www.mdpi.com/article/10.3390/app13021090/s1, Methods: Experimental methods used in this study.

**Author Contributions:** Conceptualization, J.-H.O. and S.K.K.; methodology, C.-J.J. and S.K.K.; software, J.W.K.; validation, C.-J.J., J.-S.C. and S.K.K.; formal analysis, S.K.K.; investigation, S.K.K.; resources, J.-H.O.; data curation, J.W.K.; writing—original draft preparation, J.-H.O. and J.W.K.; writing—review and editing, J.-S.C. and S.K.K.; visualization, J.W.K. and S.K.K.; supervision, J.-S.C. and S.K.K.; project administration, S.K.K.; funding acquisition, J.-H.O. All authors have read and agreed to the published version of the manuscript.

**Funding:** This research received no external funding.

**Institutional Review Board Statement:** The animal study protocol was approved by the Animal Experimentation Ethics Committee of Daegu Haany University (approval no. DHU2017-098).

**Informed Consent Statement:** Not applicable.

**Data Availability Statement:** Not applicable.

**Conflicts of Interest:** The authors declare no conflict of interest.

## Appendix A

**Table A1.** Primary antisera and detection kits used in this study.

| Antisera or Detection Kits | Code | Source | Dilution |
|---|---|---|---|
| Primary antisera * | | | |
| Anti-cleaved caspase-3 (Asp175) polyclonal antibody | 9661 | Cell Signaling Technology Inc, Beverly, MA, USA | 1:400 |
| Anti-cleaved PARP (Asp214) rat specific antibody | 9545 | Cell Signaling Technology Inc, Beverly, MA, USA | 1:100 |
| Anti-tumor necrosis factor-$\alpha$ (4E1) antibody | sc-130349 | Santa Cruz Biotechnology, Santa Cruz, CA, USA | 1:200 |
| Anti-cyclo-oxygenase (murine) polyclonal antibody | 160126 | Cayman Chemical., Ann Arbor, MI, USA | 1:200 |
| Anti-nitric oxide synthase2 (N-20) polyclonal antibody | sc-651 | Santa Cruz Biotechnology, Santa Cruz, CA, USA | 1:100 |
| Detection kits | | | |
| Vectastain Elite ABC Kit | PK-6200 | Vector Lab. Inc., Burlingame, CA, USA | 1:50 |
| Peroxidae substrate kit | SK-4100 | Vector Lab. Inc., Burlingame, CA, USA | 1:50 |

* All antisera were raised in rabbits. PARP = Cleaved poly(ADP-ribose) polymerase. ABC = Avidin-biotin-peroxidase.

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
