# Peer review of "Oral Combination Treatment of Gefitinib (IressaTM) and Sasam-Kyeongokgo: Synergistic Effects on the NCI-H520 Tumor Cell Line"

_applsci, doi:10.3390/app13021090_

Round 1

Reviewer 1 Report

This is an interesting paper on  the synergistic effects of the oral combination treatment of gefitinib (IressaTM) and sasam-2 Kyeongokgo: on NCI-H520 tumor cell line. However I would like to draw your attention to the following:

Line 75 To arrive at this conclusion you would have to use the same sasam-Kyeongogko. If so please state. 

Line 92 why female?

Line 172 Did you perform skog administration to animals with no tumor? That is did you include this significant control?

Line 148 -153 State method of determination of NK, TNF, IL...

Figure 1: effect on normal cells?===Control?

Line 197 -200 Repetition of "previous researches"

Figure 3 Please improve: provide groups in legend, align group names with pictures

Figure 5 Insert gel photo

Most importantly the References were missing in the downloaded MS 

Author Response

Reviewer 1

Comments and Suggestions for Authors

This is an interesting paper on the synergistic effects of the oral combination treatment of gefitinib (IressaTM) and sasam-Kyeongokgo: on NCI-H520 tumor cell line. However I would like to draw your attention to the following:

  1. Line 75 To arrive at this conclusion you would have to use the same sasam-Kyeongogko. If so please state.

→ According to the reviewer’s comment, we have unified to sasam-Kyeongogko in the revised manuscript.

  1. Line 92 why female?

→ Until now, lung cancer has been mainly a male disease, but recently, along with the increase in female smokers, the incidence rate in females is increasing, and it is considered an important malignant tumor in females as well. In this study, female mice were selected because they are relatively lower body weight, easy to handle, and a smaller amount of test material is required.

  1. Line 172 Did you perform skog administration to animals with no tumor? That is did you include this significant control?

→ Test material, Sasam (Adenophorae Radix) and Kyeongokgo been traditionally used in oriental medicine to treat lung damages, and there has been no toxicity-related problem to date. In accordance with the recommendation of the Animal Ethics Committee, normal mice were administered to the experimental group in order to reduce the number of animals used.

  1. Line 148 -153 State method of determination of NK, TNF, IL...

→ Currently, the manuscript is a total of 33 pages, which is not a small volume. So, we have added the methods in Supplement section.

  1. Figure 1: effect on normal cells? =Control?

→ These are the results of MTT test on normal NCI-H520 Tumor Cells.

  1. Line 197 -200 Repetition of "previous researches"

→ According to the reviewer’s comment, we have revised sentence like as “The selection, analysis, and measurement of biomarkers related to the anticancer effect and the immune-activation effect in this study were based on the methods suggested by previous researches [22-26].”

  1. Figure 3. Please improve: provide groups in legend, align group names with pictures.

→ According to the reviewer’s comment, we have added “TB = tumor-bearing; AR = Adenophorae Radix; KOG = Kyeongokgo; SKOG = Sasam-Kyeongokgo.” in the revised legend of Figure 3.

  1. Figure 5 Insert gel photo

→ Serum IL-6 and IFN-γ Levels were measured using a commercial ELISA kit, therefore, gel photos are not available. However, to make it more clear, we have added the methods in Supplement section.

  1. Most importantly the References were missing in the downloaded MS

→ Thank you for your comment. Please find the References the revised MS.

Thank you for your valuable comments on our manuscript. We have modified our manuscript according to your suggestions and we ask that the revised manuscript be re-examined and considered for publication.

Reviewer 2 Report

The authors combines gefitinib with sasam-Kyeongokgo (SKOG) for 35 days in NCI-H520 cell-bearing athymic nude mice for antitumor evaluation. In addition to tumor volume and body weight, lymphatic, periovarian fat pad weight measurements, serum level of IFN-γ,IL-6, and NK cell activity were also observed. The authors suggest co- administration of SKOG 400, 200 or 100 mg/kg with gefitinib, marked increased anti-tumor activity of gefitinib. Although the authors provided a lot of data to support combination of SKOG and gefitinib leads to synergistic effect in NCI-H520 cell-bearing athymic nude mice, several issues should be addressed before further consideration.

1.     Previous study has shown NCI-H520 is resistant to gefitinib due to low expression of EGFR (Ann Oncol. 2008 Jun;19(6):1053-9.). Clinically, the use of gefitinib would be considered as an unsuitable drug for patients with primary resistance (e.g., k-ras mutation, EGFR wt, or low EGFR expression). The authors should use the other NSCLC cell line with acquired resistance to gefitinib to show co-treatment of SKOG and gefitinib overcomes drug resistance since acquired resistance to TKIs is still inevitable and challenging.

2.     Has SKOG been tested in different cancer types before? What is the reason to evaluate antitumor activity for treatment of NSCLC?

3.     Triterpene lupeol seems to be the major component of the SKOG, the authors should demonstrate in vitro results of co-treatment of gefitinib and SKOG and gefitinib plus lupeol to make a comparison.

4.     The authors did not make a clear mark of a~d in Fig. 4~6.

5.     The authors should show the combination index (CI index) of combination group to demonstrate synergistic effect of gefitinib plus SKOG in animal model (Cancer Res (2010) 70 (2): 440–446.).

Author Response

Reviewer 2

Comments and Suggestions for Authors

The authors combines gefitinib with sasam-Kyeongokgo (SKOG) for 35 days in NCI-H520 cell-bearing athymic nude mice for antitumor evaluation. In addition to tumor volume and body weight, lymphatic, periovarian fat pad weight measurements, serum level of IFN-γ, IL-6, and NK cell activity were also observed. The authors suggest co- administration of SKOG 400, 200 or 100 mg/kg with gefitinib, marked increased anti-tumor activity of gefitinib. Although the authors provided a lot of data to support combination of SKOG and gefitinib leads to synergistic effect in NCI-H520 cell-bearing athymic nude mice, several issues should be addressed before further consideration.

  1. 1. Previous study has shown NCI-H520 is resistant to gefitinib due to low expression of EGFR (Ann Oncol. 2008 Jun;19(6):1053-9.). Clinically, the use of gefitinib would be considered as an unsuitable drug for patients with primary resistance (e.g., k-ras mutation, EGFR wt, or low EGFR expression). The authors should use the other NSCLC cell line with acquired resistance to gefitinib to show co-treatment of SKOG and gefitinib overcomes drug resistance since acquired resistance to TKIs is still inevitable and challenging.

→ In this experiment, we focused on the synergistic effect of SKOG on lung cancer, especially on the control of cachexia after chemotherapy. Currently, the most used lung cancer cell line in our laboratory is NCI-H520, so this cell line was used. Based on the current experimental results, it is judged that the synergistic effect of SKOG on lung cancer, especially NSCLC, and the possibility of controlling cachexia can be sufficiently presented. However, as suggested by the reviewer, it was judged that a similar experiment would be needed among more appropriate NSCLC cells, so the following was added as a limitation at the end of the discussion.

To clarify these points, We have added descriptions to the revised manuscript.

To the Discussion section (lines 1104-1109), the following text:

“This study has severel limitations. Since NCI-H520 cells have low expression of EGFR and related resistant to gefitinib [67], synergistic effects of SKOG and gefitinib on other NSCLC cell lines showed acquired resistance to gefitinib, along with the combination index (CI index). More detailed and abundant in vitro and in vivo experiments including efficacy tests of the major component of the SKOG, are required in future.”

To the References (lines 1283-1286), the following text:

 “ 67. Weiss, G.J.; Bemis, L.T.; Nakajima, E.; Sugita, M.; Birks, D.K.; Robinson, W.A.; Varella-Garcia, M.; Bunn, P.A. Jr; Haney, J.; Helfrich, B.A.; Kato, H.; Hirsch, F.R.; Franklin, W.A. EGFR regulation by microRNA in lung cancer: correlation with clinical response and survival to gefitinib and EGFR expression in cell lines. Ann. Oncol. 2008, 19, 1053-1059.”

  1. Has SKOG been tested in different cancer types before? What is the reason to evaluate antitumor activity for treatment of NSCLC?

→ Adenophorae Radix and Kyeongokgo, the main components of SKOG in oriental medicine, have been mainly used for the respiratory system. We have already evaluated the respiratory function improvement effects of Adenophorae Radix, KOG, SKOG. As the first attempt for anticancer activity, NSCLC cell line, which is the most common among lung cancers, was used, and since it is a natural product, combination with gefitinib, a common anticancer drug, was preferred over alone.

To clarify these points, we have added the following text:

To the Introduction section (lines 51-53), the following text:

“In addition, we have already evaluated the respiratory function improvement effects of Adenophorae Radix [19], Kyeongokgo [20] and Sasam-Kyeongokgo [21], through in vivo animal experiments.”

To the Refernces (lines 1175-1181), the following text:

  1. Hu, J.R.; Jung, C.J.; Ku, S.M.; Jung, D.H.; Ku, S.K.; Choi, J.S. Antitussive, expectorant, and anti-inflammatory effects of Adenophorae Radix powder in ICR mice. J. Ethnopharmacol. 2019, 239, 111915.
  2. Hu, J.R.; Jung, C.J.; Ku, S.M.; Jung, D.H.; Bashir, K.M.I.; Ku, S.K.; Choi, J.S. Anti-inflammatory, expectorant, and antitussive properties of Kyeongok-go in ICR mice. Pharm. Biol. 2021a, 59, 319-332.
  3. Hu, J.R.; Jung, C.J.; Ku, S.M.; Jung, D.H.; Ku, S.K.; Bashir, K.M.I.; Lee, H.J;, Choi, J.S. Deciphering the antitussive, expectorant, and anti-inflammatory potentials of ShashamKyeongok‐go and their phytochemical attributes: In vivo appraisal in ICR mice. Appl. Sci. 2021b,11, 1349.

  1. Triterpene lupeol seems to be the major component of the SKOG, the authors should demonstrate in vitroresults of co-treatment of gefitinib and SKOG and gefitinib plus lupeol to make a comparison.

→ Thanks for the great suggestion. In this experiment, we could not consider this part because we focused on the four ginseng jadeite. We will conduct further experiments with other papers in the future. Added as a limitation at the end of the discussion (lines 1104 - 1109).

  1. The authors did not make a clear mark of a~d in Fig. 4~6.

→ We have addressed the reviewer’s comment.

  1. The authors should show the combination index (CI index) of combination group to demonstrate synergistic effect of gefitinib plus SKOG in animal model (Cancer Res (2010) 70 (2): 440–446.).

→ The CI index is mainly calculated through in vitro experiments, and in the case of animal experiments, too many experimental animals are used in reality because it requires dose-dependent data for each single test substance. In this study, we focused on the synergistic anticancer effect and possible cachexia control effect according to the combination of SKOG and gefitinib using xenograft mice. Since significant increases of anticancer and cachexia control effects were detetced in the SKOG combination administration group as compared to those of gefitinib single (alone) administration group, it is considered that synergistic effects could be sufficiently shown. Therefore, unfortunately we neglected the calculation of CI index in this study. These are also added as a limitation at the end of the discussion (lines 1104 - 1109).

Thank you for your valuable comments on our manuscript. We have modified our manuscript according to your suggestions and we ask that the revised manuscript be re-examined and considered for publication.

Reviewer 3 Report

In this interesting and brilliantly written paper the authors used the traditionally applied drug ssasam in combination with the officinal drug gefitinib, and proved that their combination shows synergistic anticancer effect in comparison with treatment using gefitinib alone, so the hepatotoxic adverse effects of gefitinib can be significantly reduced by reducing the gefitinib dose and co-administering the same dose of gefitinib with sasam-Kyeongokgo.

I think that this paper should be published in the form as written.

Author Response

Reviewer 3

Comments and Suggestions for Authors

In this interesting and brilliantly written paper the authors used the traditionally applied drug ssasam in combination with the officinal drug gefitinib, and proved that their combination shows synergistic anticancer effect in comparison with treatment using gefitinib alone, so the hepatotoxic adverse effects of gefitinib can be significantly reduced by reducing the gefitinib dose and co-administering the same dose of gefitinib with sasam-Kyeongokgo.

I think that this paper should be published in the form as written.

→ Thank so much for your comments.

Round 2

Reviewer 1 Report

Comments have been successfully addressed. However experiments on control cells  do not invclude tumor cell lines but healthy cell lines e.g. stem cells

Reviewer 2 Report

Accept in present form.